# CAN WE IGNORE LABELS IN OUT-OF-DISTRIBUTION DETECTION?

**Hong Yang, Qi Yu, Travis Desell**
Rochester Institute of Technology
1 Lomb Memorial Dr, Rochester, NY 14623, USA
`{hy3134,tjdvse,qi.yu}@rit.edu`

## ABSTRACT

Out-of-distribution (OOD) detection methods have recently become more prominent, serving as a core element in safety-critical autonomous systems. One major purpose of OOD detection is to reject invalid inputs that could lead to unpredictable errors and compromise safety. Due to the cost of labeled data, recent works have investigated the feasibility of self-supervised learning (SSL) OOD detection, unlabeled OOD detection, and zero shot OOD detection. In this work, we identify a set of conditions for a theoretical guarantee of failure in unlabeled OOD detection algorithms from an information-theoretic perspective. These conditions are present in all OOD tasks dealing with real-world data: I) we provide theoretical proof of unlabeled OOD detection failure when there exists zero mutual information between the learning objective and the in-distribution labels, a.k.a. 'label blindness', II) we define a new OOD task – Adjacent OOD detection – that tests for label blindness and accounts for a previously ignored safety gap in all OOD detection benchmarks, and III) we perform experiments demonstrating that existing unlabeled OOD methods fail under conditions suggested by our label blindness theory and analyze the implications for future research in unlabeled OOD methods.

## 1 INTRODUCTION

Safety-critical applications of deep neural networks have recently become an important area of investigation in the domain of artificial intelligence, ranging from autonomous driving (Ramanagopal et al., 2018) to biometric authentication (Wang & Deng, 2021) to medical diagnosis (Bakator & Radosav, 2018). In the setting of safety-critical systems, it is no longer possible to rely on the closed-world assumption (Krizhevsky et al., 2012), where test data is drawn i.i.d. from the same distribution as the training data, known as the in-distribution (ID). These models will be deployed in an open-world scenario (Drummond & Shearer, 2006), where test samples can be out-of-distribution (OOD) and therefore should be handled with caution. OOD detection seeks to identify inputs containing a label that was never present in the training distribution. The motivation for OOD detection is simple: we do not want safety-critical systems to act on an invalid prediction, where the predicted label cannot be correct because the label was never present in training.

There is significant interest in unlabeled OOD detection due to various factors. A method that does not rely on labels can save significant costs in labeling data, as proposed by (Sehwag et al., 2021). It is also be possible to skip training on the in distribution data if such a model is generalizable, as proposed by (Wang et al., 2023). Self supervised and unlabeled learning methods can also scale to much larger datasets and it is important for these models to be robust to OOD data. Recent work in unlabeled OOD detection methods, including (Sehwag et al., 2021; Tack et al., 2020; Liu et al., 2023; Guille-Escuret et al., 2024; Wang et al., 2023), promise to improve safety using only unlabeled data. These methods can achieve even greater performance than a simple supervised baseline (Hendrycks & Gimpel, 2016), suggesting that one could replace supervised training with self-supervised learning (SSL) for a safety critical OOD detection task. This family of SSL OOD methods differ from traditional supervised OOD methods, including (Fort et al., 2021), by the use of only unlabeled data. The importance of labels is an active area of research in OOD detection (Du et al., 2024a;b).

When we view SSL from an information-theoretic perspective, the selection of features depends solely on the SSL objective and not on the labels. This, however, provides no guarantee that any features relevant for label prediction will be retained. Figure 1 provides an example of how SSL features can be less effective for identifying a label. Our theory importantly shows that, when the label-relevant features are independent of the features relevant for the SSL algorithm's successful operation, OOD detection is guaranteed to fail due to what we call 'label blindness' and that this label blindness occurs regardless of how one selects the ID dataset from the population of all data. Our experiments also suggest that Zero Shot OOD methods (Wang et al., 2023; Esmaeilpour et al., 2022) may also suffer from this issue. We show that unsupervised OOD detection methods behave in the same way as SSL in the context of information theory.

However, one can unintentionally avoid label blindness problem via the selection of the OOD dataset when constructing OOD benchmarks. Existing methods generally consider ID and OOD data from different datasets, e.g., (Fort et al., 2021), (Sehwag et al., 2021), and (Hendrycks et al., 2019). In these benchmarks, there is no significant overlap between the ID and OOD input data, allowing OOD detection algorithms to succeed on features independent of the label. To address this issue and to test for label blindness, we introduce the Adjacent OOD detection task to evaluate the performance on OOD detection algorithms when there is significant overlap between the OOD input data and ID input data. We also prove that it is impossible to guarantee that a real world system will never encounter OOD input data that significantly overlaps ID input data.

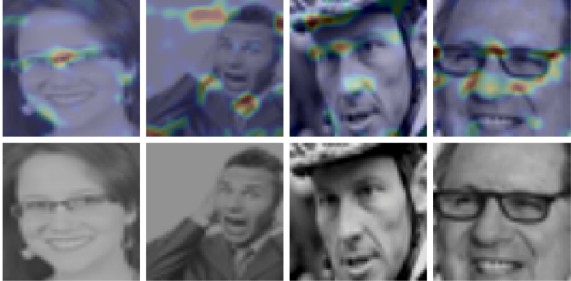

Figure 1: An example failure case by visualizing the heatmaps of the gradient of a unlabeled SimCLR trained Resnet (Chen et al., 2020) using the GradCAM method (Selvaraju et al., 2017). The OOD detection task is to detect OOD facial expressions. In this case, the OOD detection method fails as justified by our theoretical work, where the representations do not exhibit a strong gradient in regions commonly associated with facial expressions (i.e., eyebrows, mouth, etc.).

This work aims to answer the following question: *can we ignore labels when engaging in OOD detection?* Through numerous experiments and theoretical proofs, we show that it is not safe to ignore labels when performing OOD detection. This is contrary to the increasing recent efforts that propose new self supervised, unsupervised, and other unlabeled OOD detection methods. This work's key contributions include:

- **The Label Blindness Theorem.** We theoretically prove that any SSL or Unsupervised Learning algorithm will fail when its information required for the surrogate task is independent of the information required for predicting labels. Through this proof, we conclude that there cannot be a generally applicable SSL or Unsupervised learning OOD detection algorithm as there will always exist independent labels due to the no free generalization theorem, see theorem 3.5.
- **Adjacent OOD detection benchmarks.** We introduce the concept of bootstrapping without replacement of the ID labels to create the Adjacent OOD detection task. To the authors' knowledge, this OOD detection task is novel to and absent from research in OOD detection. This task evaluates OOD detection when there is significant overlap in OOD data and ID. We also theoretically prove that overlapping OOD and ID data is possible in every real world dataset.
- **Impact on existing and future OOD methods.** We demonstrate that existing SSL and Unsupervised Learning OOD methods fail under the conditions suggested by our theory and that existing benchmarks do not capture such failures. We also evaluate zero shot OOD detection methods, which fail in a similar manner to SSL and Unsupervised Learning OOD methods. We make recommendations on the development and testing of future OOD methods.

## 2 PRELIMINARIES

### 2.1 LABELED AND UNLABELED OUT-OF-DISTRIBUTION DETECTION

The task of out-of-distribution detection is to identify a semantic shift in the data (Yang et al., 2021). This is determining when no predicted label could match the true label $\boldsymbol{y} \notin \mathbb{Y}_{in}$, where $\mathbb{Y}_{in}$

represents the set of in-distribution training labels. In this case, we would consider the semantic space of the sample and the training distribution to be different, representing a semantic shift. We can express the probability that a sample is out-of-distribution via $P(\boldsymbol{y} \notin \mathbb{Y}_{in}|\boldsymbol{x})$. One baseline method to calculate $P(\boldsymbol{y} \notin \mathbb{Y}_{in}|\boldsymbol{x})$ is to take $1 - \texttt{MSP}(\boldsymbol{x})$, where $\texttt{MSP}$ is the maximum softmax probability from a classifier for a particular datapoint.

Furthermore, we are only concerned with labels that can be generated using only $\boldsymbol{x}$, via function $f$ which depends solely on $\boldsymbol{x}$ and no other information. $f$ may represent human labelers that generate $\boldsymbol{y}$. If we consider $\mathbb{Y}_{all}$ as the set of all possible labels that can be generated from $f(\boldsymbol{x} \in \mathbb{X}_{all})$, a subset of $\mathbb{X}_{all}$ considered as $\mathbb{X}_{training}$ may not contain all labels in $\mathbb{Y}_{all}$. For real world datasets, it is possible that $\mathbb{Y}_{in} \subsetneq \mathbb{Y}_{all}$.

We can also approach the problem of OOD detection without the use of labels. One can train a model on ID data using a surrogate task for the purposes of computing a metric. For example, (Sehwag et al., 2021) trains a resnet with SimCLR and computes the Mahalanobis distance between the training representations and the test sample representations to compute the OOD score. Alternatively, one could utilize a pretrained model with broad knowledge to compute a metric to use as the OOD score, such as in (Wang et al., 2023).

## 2.2 SELF-SUPERVISED AND UNSUPERVISED LEARNING

This section covers representation learning and its implications for SSL and unsupervised learning. If there is no mutual information between two random variables, neither can be used to reduce uncertainty about the other (Shannon, 1948). In both self-supervised and unsupervised OOD detection, if there is no mutual information between the intermediate representations and the OOD detection task, the OOD detection system cannot reduce uncertainty with respect to the OOD detection task using the intermediate representations.

Representation learning can be formulated as finding a distribution $p(\mathbf{z}|\mathbf{x})$ that maps the observations from $\boldsymbol{x} \in \mathbb{X}$ to $\boldsymbol{z} \in \mathbb{Z}$, while capturing relevant information for some primary task. When $\mathbf{y}$ represents some primary task, we consider only $\mathbf{z}$ that is sufficiently discriminative for accomplishing the task $\mathbf{y}$. For simplicity, we consider $\mathbf{y}$ as a classification label, but $\mathbf{y}$ can represent any objective or task. Federici et al. (2020) show that this sufficiency is met when the information relevant for predicting $\mathbf{y}$ is unchanged when encoding $\mathbf{x} \rightarrow \mathbf{z}$.

**Definition 2.1.** Sufficiency: A representation $\mathbf{z}$ of $\mathbf{x}$ is sufficient for $\mathbf{y}$ if and only if $I(\mathbf{x}; \mathbf{y} \mid \mathbf{z}) = 0$.

Since there exists the sufficient statistic $\mathbf{x} = \mathbf{z}$, we must consider the minimal sufficient statistic which conveys only relevant information for predicting $\mathbf{y}$. An SSL algorithm seeks to learn the minimal sufficient statistic via the information bottleneck framework (Shwartz-Ziv & LeCun, 2023).

**Definition 2.2.** Minimal Sufficient Statistic. A sufficient statistic $\mathbf{z}$ is minimal if, for any other sufficient statistic $\mathbf{s}$, there exists a function $f$ such that $\mathbf{z} = f(\mathbf{s})$.

Information bottleneck optimization can be expressed as the minimization of the representation's complexity via $I(\mathbf{x}; \mathbf{z})$ and maximizing its utility $I(\mathbf{z}; \mathbf{y})$. This results in the information theoretic loss function below, where $\beta$ is a trade-off between complexity and utility (Shwartz-Ziv & LeCun, 2023). In practice, learning $\mathbf{z}$ without $\mathbf{y}$ requires a surrogate task $\mathbf{y}_s$, e.g., (Chen et al., 2020), with the loss defined as:

$$\mathcal{L} = I(\mathbf{x}; \mathbf{z}) - \beta I(\mathbf{z}; \mathbf{y}). \tag{1}$$

It should be noted that the primary task $\mathbf{y}$ may be equal to the SSL task $\mathbf{y}_s$. In such a case, compression towards the minimal sufficient statistic still occurs. This is important because unsupervised methods for deep neural networks (DNNs) will use a surrogate task $\mathbf{y}_u$ to train the DNN's weights. Thus, if we assign the primary task for an unsupervised learning method to be equal to its surrogate task, it will behave identically to SSL from the perspective of information theory.

When $\mathbf{x}$ has higher information content than $\mathbf{y}$, there exists information in $\mathbf{x}$ that is not relevant for predicting $\mathbf{y}$. This can be better understood by dividing $I(\mathbf{x}; \mathbf{z})$ into two terms (Federici et al., 2020) as follows:

$$I(\mathbf{x}; \mathbf{z}) = \underbrace{I(\mathbf{x}; \mathbf{z} \mid \mathbf{y})}_{\text{superfluous information}} + \underbrace{I(\mathbf{z}; \mathbf{y})}_{\text{predictive information}} . \tag{2}$$

However, superfluous information is not affected by the labels of primary task, only by $\mathbf{x}$ and $\mathbf{y}_s$. Using information theory, we can show that any SSL OOD detection algorithm will fail when the surrogate task $\mathbf{y}_s$ is independent of the labels in the in-distribution dataset. This applies to unsupervised OOD detection algorithms that also use a surrogate task.

## 3 GUARANTEED OOD DETECTION FAILURE

This section introduces the concept of **Label Blindness**, with one key supporting theorem and one key supporting lemma. The full proofs for these theoretical results are provided in Appendix D. Note that $R_{\mathbf{x}}$ represents the support of random variable $\mathbf{x}$ such that $R_{\mathbf{x}} = \{\boldsymbol{x} \in \mathbb{R} : P(\boldsymbol{x}) > 0\}$. For clarity, we refer to cases where $I(\mathbf{x}_1; \mathbf{x}_2) = 0$ as **Strict Label Blindness** and discuss **Approximate Label Blindness** $I(\mathbf{x}_1; \mathbf{x}_2) \approx 0$ at the end of section 3.1.

### 3.1 LABEL BLINDNESS THEOREM (STRICT LABEL BLINDNESS)

We identify a guarantee of OOD detection failure for any information bottleneck-based optimization process if the unlabeled learning objective is independent from labels used to determine the ID set, described by Corollary 3.3. This corollary is derived from two concepts: strict label blindness in the minimal sufficient statistic and the independence of filtered distributions. We first consider the minimal sufficient statistic and how it leads to strict label blindness; see Theorem 3.1.

**Theorem 3.1.** *Strict Label Blindness in the Minimal Sufficient Statistic.*
*Let $\mathbf{x}$ come from a distribution. $\mathbf{x}$ is composed of two independent variables $\mathbf{x}_1$ and $\mathbf{x}_2$. Let $\mathbf{y}_1$ be a surrogate task such that $H(\mathbf{y}_1|\mathbf{x}_1) = 0$. Let $\mathbf{z}$ be any sufficient representation of $\mathbf{x}$ for $\mathbf{y}_1$ that satisfies the sufficiency definition 2.1 and minimizes the loss function $\mathcal{L} = I(\mathbf{x}_1\mathbf{x}_2; \mathbf{z}) - \beta I(\mathbf{z}; \mathbf{y}_1)$. The possible $\mathbf{z}$ that minimizes $\mathcal{L}$ and is sufficient must meet the condition $I(\mathbf{x}_2; \mathbf{z}) = 0$.*

*Detailed proof in Appendix D.4.*

Intuitively, the minimal sufficient representation cannot encode any information independent of the surrogate learning objective, otherwise it would not be minimal. This means that the representation will be blind to any label built upon the independent information.

However, Theorem 3.1 is not sufficient to guarantee OOD failure. This is because the selection of the ID training set could change the learned representation $\mathbf{z}$, possibly improving OOD detection performance by increasing mutual information, $I(\mathbf{x}_2; \mathbf{z}) > 0$. We formally disprove this possibility through Lemma 3.2.

**Lemma 3.2.** *Independence of Filtered Distributions.*
*Let $\mathbf{x}$ come from a distribution. $\mathbf{x}$ is composed of two independent variables $\mathbf{x}_1$ and $\mathbf{x}_2$. For $\mathbf{x}_2'$ where $R_{\boldsymbol{x}_2'} \subset R_{\boldsymbol{x}_2}$, there exists no $\mathbf{x}_2'$ such that $H(\mathbf{x}_1|\mathbf{x}_2') < H(\mathbf{x}_1)$.*

*Detailed proof in Appendix D.5.*

Lemma 3.2 states that filtering on a label generated on one of two independent variables cannot provide information about the other. This applies to the selection of ID data from the population, if the selection criteria is independent of the learning objective. This means that the strict label blindness properties predicted by Theorem 3.1 will apply to ID training data. These two concepts bring us to our main result – strict label blindness in filtered distributions; see Corollary 3.3.

**Corollary 3.3.** *Strict Label Blindness in Filtered Distributions.*
*Let $\mathbf{x}$ come from a distribution. $\mathbf{x}$ is composed of two independent variables $\mathbf{x}_1$ and $\mathbf{x}_2$. Let $\mathbf{y}_1$ be a a surrogate task such generated by $\boldsymbol{y}_1 = f_1(\boldsymbol{x}_1)$ $H(\mathbf{y}_1|\mathbf{x}_1) = 0$. Let $\mathbf{y}_2$ be a label such that $H(\mathbf{y}_2|\mathbf{x}_2) = 0$ and $\boldsymbol{y}_2 = f_2(\boldsymbol{x}_2)$. Let $\mathbb{Y}_{in}$ be as subset of labels $\mathbb{Y}_{in} \subset R_{\mathbf{y}_2}$. Let $\mathbf{x}'$ be a subset of $\mathbf{x}$ where $R_{\mathbf{x}'} = R_{\mathbf{x}} \cap \{\boldsymbol{x} \in \mathbb{R} : f_2(\boldsymbol{x}_2) \in \mathbb{Y}_{in}\}$ such that $\mathbf{x}'$ is composed of independent variables $\mathbf{x}_1'$ and $\mathbf{x}_2'$ and $\boldsymbol{y}_1' = f_1(\boldsymbol{x}_1')$. The sufficient representation $\mathbf{z}$ learned by minimizing $\mathcal{L} = I(\mathbf{x}_1'\mathbf{x}_2'; \mathbf{z}) - \beta I(\mathbf{z}; \mathbf{y}_1')$ must have $I(\mathbf{x}_2; \mathbf{z}) = 0$ and $I(\mathbf{y}_2; \mathbf{z}) = 0$.*

*Detailed proof in Appendix D.6.*

This means that, when we select the ID training data, if the selection criteria and labels are independent of the surrogate learning objective, then we can guarantee failure in OOD detection due to the absence of any information in the learned representation $\mathbf{z}$. For simplicity, we refer to this concept, supported by Corollary 3.3, as the problem of **Strict Label Blindness**.

In summary, when the surrogate learning task can be achieved without learning about features relevant for the label, it will not learn any features relevant for the label. Figure 1 is a visualized example of this. If the SSL or unsupervised learning method fails to learn any label-relevant features, then any OOD detection algorithm built from those representations cannot differentiate between the labels selected as ID and those not selected as ID. This guarantees failure in OOD detection because no label information passes through the information bottleneck.

**Implications of Strict Label Blindness in Real World Situations**    We can utilize Fano Inequality to extend our understanding of strict label blindness to consider situations of where the variables are not fully independent. The lower bound for prediction error is defined by the entropy of the target label $\mathbf{y}$ less the mutual information between the input $\mathbf{x}$ and target label, as shown in Theorem 3.4. Under strict label blindness, when $I(\mathbf{x}; \mathbf{y}) = 0$, the lower bound for error is at its maximum. When $I(\mathbf{x}; \mathbf{y}) \approx 0$, the lower bound for error is large enough to be unreliable. We refer to this condition as **Approximate Label Blindness** and we conduct experiments to evaluate this condition. Unless specified as strict, label blindness refers to the approximate case.

**Theorem 3.4.** *Fano's Inequality (See (Robert, 1952)).*
*Let $\mathbf{y}$ be a discrete random variable representing the true label with $\mathcal{Y}$ possible values and cardinality of $|\mathcal{Y}|$ and $\mathbf{x}$ be a random variable used to predict $\mathbf{y}$. Let $e$ be the occurrence of an error such that $\mathbf{y} \neq \hat{\mathbf{y}}$ where $\hat{\mathbf{y}} = f(\mathbf{x})$. Let $H_b$ represent the binary entropy function such that $H_b(e) = -P(e) \log P(e) - (1 - P(e)) \log(1 - P(e))$. The lower bound for $P(e)$ increases with lower $I(\mathbf{x}; \mathbf{y})$.*

$$H_b(e) + P(e) \log(|\mathcal{Y}| - 1) \geq H(\mathbf{y}) - I(\mathbf{x}; \mathbf{y}). \tag{3}$$

## 3.2 Distinctions from the State-of-the-Art

Previous work by (Federici et al., 2020) introduced the *No Free Generalization Theorem*, which contains some similar concepts. It states that a compressed representation of $\mathbf{x}$ cannot contain information for all possible labels of $\mathbf{x}$, but does not guarantee OOD detection failure.

**Theorem 3.5.** *No Free Generalization (See (Federici et al., 2020)).*
*Let $\mathbf{x}, \mathbf{z}$ and $\mathbf{y}$ be random variables with joint distribution $p(\mathbf{x}, \mathbf{y}, \mathbf{z})$. Let $\mathbf{z}'$ be a representation of $\mathbf{x}$ that satisfies $I(\mathbf{x}; \mathbf{z}) > I(\mathbf{x}; \mathbf{z}')$, then it is always possible to find a label $\mathbf{y}$ for which $\mathbf{z}'$ is not predictive for $\mathbf{y}$ while $\mathbf{z}$ is.*

Our theoretical work shows the exact conditions that guarantee OOD failure based on the mutual information between the labels and the loss function, when performing SSL or unsupervised learning.

Recent work by (Du et al., 2024b) investigates how labels can improve the performance of OOD algorithms. Their work does not describe any guarantee of failure in OOD detection in the absence of labels, but does support the idea that labels are important for OOD detection.

## 3.3 Theoretical Implications

Our work applies to deep neural networks (DNNs) trained without labels for the purpose of OOD detection. The key assumption of information bottleneck compression is generally applicable to DNNs (Shwartz-Ziv & Tishby, 2017). Regardless of other assumptions, such as the multi-view assumption, an information bottleneck DNN trained without labels will still compress data irrelevant to its loss objective, even if that data is relevant for its intended task. It does not matter what task the training process was originally designed for because the unlabeled training process ultimately generates/adheres to its own learning objective. For any learning objective, there will exist an independent feature unless compression is not possible, as in $I(\mathbf{x}; \mathbf{y}) = I(\mathbf{x}; \mathbf{x})$. If there is compression, then there exists labels for which OOD detection failure is guaranteed.

Our work predicts a guarantee of failure only when we consider the OOD set of all non-ID data. In our own experiments and in work by (Sehwag et al., 2021; Hendrycks et al., 2019; Liu et al., 2023), purely self-supervised and unsupervised OOD methods can perform well against common benchmark OOD sets. This suggests that the choice of ID set and OOD set pairs can unintentionally hide label blindness failure. Alternatively, we can also construct a test to identify if the OOD detection algorithm suffers from label blindness. To construct such a test, we rely on the insight from Corollary 3.3 and the use of a simple statistical method.

# 4 BENCHMARKING FOR LABEL BLINDNESS FAILURE

## 4.1 BOOTSTRAPPING AND THE ADJACENT OOD BENCHMARK

One logical consequence of Corollary 3.3 is that one cannot avoid failure due to label blindness by selecting different labels for one's ID set, so long as the label selection is independent from the **blue**learning objective. To test any OOD detection algorithm for label blindness failure, this simply entails selecting different labels for one's ID set. To construct this benchmark, we randomly sample labels to be considered as ID and other labels to be consider OOD. This is similar to bootstrapping, but without replacement. If an OOD detection algorithm is 'approximately label blind', its average OOD detection performance across the samples should be poor. We refer to this as the **Adjacent OOD Detection Benchmark**.

## 4.2 WHY ADJACENT OOD IS SAFETY-CRITICAL TO ALMOST ALL REAL WORLD SYSTEMS

The Adjacent OOD detection benchmark evaluates the performance of OOD detection algorithms when there may be a significant overlap between the ID data and OOD data. This condition applies to all systems where it is impossible to guarantee that there will be no significant overlap in the feature space between ID and OOD data. This is true for almost all real world systems and is theoretically proven below.

**Theorem 4.1.** *Unavoidable Risk of Overlapping OOD Data*

*Let $\mathbf{x}$ come from a distribution. Let $f$ be some labeling function to generate labels $\mathbf{y}$ such that $\boldsymbol{y} = f(\boldsymbol{x})$, where there are at least two unique labels $|R_{\mathbf{y}}| > 1$. Let $\mathbf{x}_{in}$ be a random subset of $\mathbf{x}$ where $R_{\mathbf{x}_{in}} \subsetneq R_{\mathbf{x}}$ and $|R_{\mathbf{x}_{in}}| < \infty$. Let $\mathbf{y}_{in}$ be labels generated from $\boldsymbol{y}_{in} = f(\boldsymbol{x}_{in})$. The probability that a randomly selected $\boldsymbol{x}$ contains $\boldsymbol{y}$ not present in $R_{\mathbf{y}_{in}}$ is always greater than 0. Detailed proof in Appendix D.7.*

In theory, this risk can be reduced to an acceptable level by adding more data to the training dataset. However, this reduction in risk requires the assumption that the collected data is randomly sampled. This is almost never true for real world datasets and often the opposite is true, where the nature of sampling can significantly increase this risk.

One risk factor present in every real world dataset is the dataset creation date. By creating the dataset at any specific point in time, the dataset cannot be randomly sampled with respect to time because it is impossible to collect data from the future. For example, if one where to create a dataset of diseases today, it would not contain any future diseases. In this example, the probability that the training dataset is incomplete is 100%, which guarantees that there will be OOD data that significantly overlaps with ID data. For most real world systems, the only safe assumption is that there may be OOD data that overlaps with ID data and it is necessary to plan accordingly.

The failure predicted by the label blindness theory is easiest to detect in the adjacent OOD situation. Where there is a likelihood of adjacent data, Theorem3.3 predicts OOD detection failure. Where there is no adjacent data, features independent of the label can still be used to distinguish between ID data and non adjacent OOD data, as shown in various experiments in this paper and others (Sehwag et al., 2021; Hendrycks et al., 2019; Liu et al., 2023).

## 4.3 COMPARING ADJACENT, NEAR, AND FAR OOD BENCHMARKS

Many unlabeled OOD methods generally perform quite well on far and near OOD tasks. Far OOD is often defined by ID and OOD sets with different semantic labels and styles (Fang et al., 2022). One such far OOD benchmark is MNIST as ID data and CIFAR10 as OOD data. Near OOD contains ID and OOD sets with similar semantic labels and styles (Fang et al., 2022). These tasks tend to be more difficult for existing OOD detection methods than far OOD detection tasks. One such near OOD benchmark is CIFAR10 as ID and CIFAR100 as OOD. However, the overlap in the near OOD detection benchmarks is significantly less than the adjacent OOD detection benchmark, which evaluates the maximum possible feature overlap. For example, an Adjacent OOD benchmark on the ICML Facial Expressions dataset may contain the same face with different expressions, resulting in significant feature overlap. These existing benchmarks do not provide sufficient safety guarantees in applications where there may be significant overlap between ID and OOD data.

### 4.4 IMPLICATIONS FOR OOD FROM UNLABELED DATA

While methods that utilize only unlabeled data, such as (Sehwag et al., 2021; Liu et al., 2023; Guille-Escuret et al., 2024), show promising results on both near and far OOD tasks, their performance in the adjacent OOD detection tasks depends on the mutual information between the learned representation and the ID labels. Our theoretical work suggests that such methods will perform poorly, if the surrogate task is independent of the labels.

The adjacent OOD detection benchmark can also evaluate the performance of zero shot OOD detection methods. While our theoretical work does not extend to pretraining due to the use of labels, it is also still important to consider the performance when OOD data overlaps ID data.

## 5 EXPERIMENTAL RESULTS

We conduct the following experiments to verify the existence of label blindness in unlabeled OOD detection methods. All hyperparameters and configurations were the best performing from their respective original paper implementations, unless noted otherwise. Experiments are repeated 3 times. Note that code for fully replicating experiments of this work can be found at `https://github.com/hyang0129/ProblematicSelfSupervisedOOD`

### 5.1 EXPERIMENTAL SETUP

**Supervised Baseline.** We use Maximum Softmax Probability (MSP) (Hendrycks & Gimpel, 2016) as our baseline supervised method for comparison. We augment the training data using random rotation, horizontal flip, random crop, gray scale, and color jitter. Images are resized to $64 \times 64$. We train using stochastic gradient descent with momentum and a cosine annealing learning schedule. We train for 10 warm up epochs followed by 150 regular epochs, selecting the weights with the highest validation accuracy. We use a standard ResNet50 architecture.

**Self-supervised Baselines.** We use two SSL methods to evaluate how representations are learned, SimCLR (Chen et al., 2020) and Rotation Loss (RotLoss) (Hendrycks et al., 2019). Images are resized to $64 \times 64$ for both cases. For SimCLR, we augment the training data using random rotation, horizontal flip, random crop, gray scale, and color jitter. For Rotation Loss, we use only random crop and horizontal flip. We train using stochastic gradient descent with momentum (and a cosine annealing learning schedule) and employ a standard ResNet50 architecture and train for 10 warm up epochs followed by 500 regular epochs, selecting the weights with the best-learned representations. We use a KNN classifier to determine the best representations during validation at the end of each epoch.

To evaluate OOD performance, we use two methods to generate the OOD score of each sample, SSD (Sehwag et al., 2021) and KNN, similar to (Sun et al., 2022). SSD considers the OOD score as the Mahalanobis distance of the sample from the center of all in-distribution training data samples. The KNN method considers the OOD score as the Euclidean distance from the $N$th nearest neighbor of the test sample to all in-distribution training samples. Both methods are distance based OOD detection and are commonly used with representation learning. We use the same representation mentioned in the previous paragraph.

**Unsupervised Baseline.** To consider how an unsupervised OOD detection method functions, we evaluate the diffusion impainting OOD detection method proposed by (Liu et al., 2023) using code provided in their paper's linked repository. We utilize the training configuration that generated the paper's main results, which involved an alternating checkerboard mask $8 \times 8$, an LPIPS distance metric to calculate the OOD score, and 10 reconstructions per image. We modify only the input image size to be $64 \times 64$ for all datasets and run additional experiments to evaluate performance on their alternative MSE distance metric. This method is representative of other generative methods, such as Xiao et al. (2020).

**Zero-shot Baseline.** To consider how well zero shot learning algorithms perform, we evaluate the CLIPN model presented by (Wang et al., 2023). We utilize their pretrained weights provided in their paper's repository and perform zero shot OOD detection on our adjacent OOD detection benchmark. We evaluate CLIPNs performance using 3 of their paper's algorithms, Maximum Softmax Probability, Compete to Win (CTW), and Agree to Differ (ATD).

## 5.2 ADJACENT OOD DATASETS

To create the Adjacent OOD detection task, we randomly split 25% of all classes into the OOD set and retain 75% as the ID set. We also repeat our experiments three times with different seeds to account for different splits of the ID and OOD set. Only ICML Facial expressions has a major class imbalance for one of its seven classes. See Appendix E for examples of the datasets.

The ICML Facial Expressions dataset (Erhan et al., 2013) contains seven facial expressions split across $28,709$ faces in the train set and $7,178$ in the test set. The expressions include anger, disgust, fear, happiness, sadness, surprise, and neutral. Self-supervised algorithms may not learn relevant features for distinguishing expressions and instead learn features relevant for distinguishing faces.

The Stanford Cars dataset (Krause et al., 2013) contains $16,185$ images taken from 196 classes of cars. The data is split into $8,144$ training images and $8,041$ testing images, with each class being split roughly 50-50. Classes are typically very fine-grained, at the level of Make, Model, Year, e.g., 2012 Tesla Model S or 2012 BMW M3 coupe. This creates a particularly challenging Adjacent OOD task because of the reliance on more subtle features to differentiate cars.

The Food 101 dataset by (Bossard et al., 2014) consists of 101 food categories and $101,000$ images. There are 250 manually reviewed test images and 750 training images for each class. Note that training images were not cleaned to the same standard as the test images and will contain some mislabeled samples. We believe that this should not significantly detract from the Adjacent OOD nature of the dataset.

## 5.3 EXPERIMENTAL RESULTS

Experimental results for Adjacent OOD are presented in Table 1. It is apparent that the baseline supervised method performs better than most unlabeled methodss on the Adjacent OOD detection task. In cases where the unlabeled methods exhibits performance as good as random guessing, it is likely that the learned representation contains little information about the semantic label. This is contrary to the reported performance improvements presented in unlabeled OOD papers (Sehwag et al., 2021; Hendrycks et al., 2019; Liu et al., 2023), as our experimental results suggest unlabeled OOD is significantly worse than a simple MSP baseline.

It is important to note that the zero shot CLIPN method performs well when the label text's usage in pretraining is similar to the label text's usage in the ID data. In the case of the Cars dataset, the pretraining dataset CC3M (Sharma et al., 2018) contains many images captioned with the make and model of various cars, resulting in good performance. The Food dataset also sees similar label usage in the pretraining set. However, the Faces dataset's labels are not aligned. For example, there are multiple images associated with the emotion angry that do not contain a human face, such as an image of a angry fist. When there is little or no mutual information between the pretraining data and the ID labels, zero shot methods will perform poorly in OOD detection tasks. Examples of pretraining data are provided in the appendix G.

In appendix F.1, we show adjacent OOD results for CIFAR10 and CIFAR100. We observe decent OOD performance on the unlabeled SimCLR compared to the labeled supervised MSP. This is likely because the SimCLR algorithm is better at learning the relevant features in these datasets and that the classes are more visually dissimilar, resulting in less overlap of OOD and ID data. In appendix F.2 we show strong results far OOD performance for SimCLR based OOD detection, which confirms findings in papers that test unlabled OOD methods against a far OOD detection benchmark, (Sehwag et al., 2021; Tack et al., 2020; Liu et al., 2023; Guille-Escuret et al., 2024; Wang et al., 2023).

## 6 RELATED WORK

**Out-of-Distribution Detection.** (Yang et al., 2021) defines OOD detection as the detection of a semantic shift. A semantic shift is a shift in the label space, where the label for a sample does not exist in the training set. OOD detection is crucial in applications where failure is very costly and/or the probability of OOD inputs are very high, as in autonomous driving (Huang et al., 2020). Following (Hendrycks & Gimpel, 2016), there have been many advances in supervised OOD detection. Some of these improved methods include those based on the ODIN score (Liang et al., 2017), Mahalanobis distance (Lee et al., 2018), energy score (Liu et al., 2020), minimum other score (Huang & Li, 2021), and deep Adjacent-neighbors (Sun et al., 2022). These methods differ from self-supervised methods through the use of label information during training.

Table 1: Results from experiments across various datasets and methods. Unlabeled methods perform poorly in adjacent OOD detection. CLIPN performance is due to labels present in the pretraining dataset and is discussed in section 5.3. Higher AUROC and lower FPR is better.

| Method | Faces | | Cars | | Food | |
|---|---|---|---|---|---|---|
| | AUROC | FPR95 | AUROC | FPR95 | AUROC | FPR95 |
| Supervised MSP | 70.8±0.3 | 88.2±0.2 | 69.2±0.9 | 88.8±0.8 | 78.8±1.2 | 81.1±1.6 |
| SimCLR KNN | 52.0±4.2 | 95.0±1.3 | 52.5±0.4 | 94.0±0.5 | 61.1±2.8 | 91.6±1.6 |
| SimCLR SSD | 55.0±4.5 | 95.1±2.0 | 52.7±0.7 | 93.7±1.1 | 64.4±0.8 | 89.3±0.5 |
| RotLoss KNN | 46.1±2.5 | 95.8±0.4 | 51.1±0.6 | 94.8±0.7 | 49.7±3.8 | 94.9±0.9 |
| RotLoss SSD | 46.6±3.0 | 95.7±0.5 | 50.7±1.9 | 95.0±1.2 | 50.7±3.6 | 94.9±0.9 |
| Diffusion LPIPS | 54.7±4.6 | 94.2±3.7 | 53.8±1.8 | 93.9±1.2 | 52.9±2.2 | 94.4±0.6 |
| Diffusion MSE | 55.3±2.2 | 94.2±1.4 | 51.6±1.6 | 94.4±0.5 | 52.5±3.4 | 94.2±0.6 |
| CLIPN CTW | 47.0±1.4 | 97.3±0.3 | 65.0±5.1 | 69.4±9.4 | 70.9±2.9 | 69.1±7.0 |
| CLIPN ATD | 44.2±1.4 | 97.5±0.2 | 81.1±4.3 | 56.6±10.4 | 84.9±0.2 | 53.9±4.5 |
| CLIPN MSP | 58.7±4.4 | 95.9±1.4 | 76.5±1.4 | 75.4±0.6 | 80.5±1.6 | 74.0±1.4 |

**Self-Supervised and Unsupervised Learning.**  Unsupervised learning involves finding underlying patterns within unlabeled data. Diffusion models (Ho et al., 2020) and generative adversarial networks (GAN) (Karras, 2017) can be considered as unsupervised learning. Most DNN unsupervised learning methods define some task based learning objective, such as the reconstruction task via autoencoders (Baldi, 2012) or the discrimination between real and fake data in GANs (Creswell et al., 2018).

SSL can be considered a variation of unsupervised learning that focuses on learning a representation $Z$ from input $X$, with respect to task $Y$, such that $I(Z;Y)$ is maximized and $I(X;Z)$ is minimized. The unsupervised methods referenced in previous paragraph can be considered SSL with respect to their learning objectives. Notably, (Shwartz-Ziv & LeCun, 2023) provides a unified information-theoretic view of SSL. Recent advances in SSL methodology include joint embeddings based on SimCLR (Chen et al., 2020) and SimSiam (Chen & He, 2021), as well as those based on generative models (He et al., 2022).

**Unlabeled Out-of-Distribution Detection.**  Unlabeled OOD detection methods include self supervised and unsupervised methods. Self-supervised OOD detection methods can vary significantly in their definition of the term self-supervised. Methods that train on OOD information (Mohseni et al., 2020) are inherently biased towards better performance on the trained OOD sets. Methods accessing in-distribution labels (Vyas et al., 2018) are not self-supervised. We consider the self-supervised OOD detection as any OOD method that does not access in-distribution labels nor the out-of-distribution set. With this definition, all self-supervised OOD methods must contain an SSL objective and some way to determine the OOD score based on model output, which is often times a metric based on the model's learned representation(s). Many unsupervised OOD detection methods fall under this definition as well, such as (Daxberger & Hernández-Lobato, 2019; Liu et al., 2023; Xiao et al., 2020).

Methods based on representation and scoring were presented in the following key efforts. (Hendrycks et al., 2019) combines a rotation loss and rotation score whereas (Sehwag et al., 2021) utilizes SimCLR (Chen et al., 2020) and the Mahalanobis distance. (Khalid et al., 2022) combines adversarial contrastive learning with singular value decomposition while (Zhang et al., 2021), in contrast, combines flow-based generated models with the Kolmogorov–Smirnov distance. The method of (Xiao et al., 2020) is based on integration of variational auto encoders with likelihood regret.

Contrastive representation learning methods have been used to improve the robustness of supervised OOD detectors. (Sun et al., 2022) uses supervised contrastive learning (Khosla et al., 2020) to improve the performance of a KNN-based OOD detector. Most recently, work by (Guille-Escuret et al., 2024) utilizes maximum mean discrepancy combined with contrastive SSL.

Zero shot OOD detection methods are a more recent development in unlabeled OOD detection. These methods utilize the CLIP model presented in (Radford et al., 2021) with two notable and recent publications (Esmaeilpour et al., 2022) and (Wang et al., 2023).

## 7 DISCUSSION

### 7.1 IMPACT OF LABEL BLINDNESS ON FUTURE RESEARCH

A consequence of the label blindness theorem is that there cannot exist a single unlabeled OOD detection algorithm for all unlabeled data. However, unlabeled learning methods, such as SimCLR, are vital for improving OOD detection. The model of (Sun et al., 2022) learns representations using a supervised version of SimCLR, similar to (Khosla et al., 2020). The combination of a multi-view information bottleneck with supervised classes produces a more robust representation of the in-distribution data than using only a supervised loss. Recent work by (Du et al., 2024a) provides a strong theoretical basis for why unlabeled data can improve OOD detection performance.

Future work should focus on overcoming approximate label blindness through few or one shot methods that can better incorporate label information. Such methods could incorporate a tiny amount of labeled data to avoid the complete independence condition described in Theorem3.3. More work needs to be done to determine how much label information is sufficient for the adjacent OOD detection detection benchmark.

### 7.2 IMPACT OF LABEL BLINDNESS ON REAL WORLD PROBLEMS

One can still enjoy the benefits of unlabeled OOD methods when the consequences of label blindness are acceptable. For example, if the objective was to detect any disease at all, then detecting a novel disease as in distribution would not be problematic. Unlabeled OOD can also be used in cases where the risk of adjacent OOD data is acceptably low. The risk defined by Theorem 4.1 is less relevant when randomness in the collection of ID data can be ensured. For example, an ID dataset of World War 2 aircraft would not be biased by the collection date and the risk of overlapping OOD data can be reduced to effectively zero.

Unlabeled OOD detection can also work well when the learned features are relevant for the OOD setting. In the case of adjacent OOD detection, an unlabeled method should perform well if the learning objective is closely related to the ID labels. Alternatively, unlabeled OOD detection can be in used cases where one expects only near and far OOD data.

## 8 CONCLUSION

In this work we provide an answer to the question, can we ignore labels for OOD detection? Our theoretical work shows that the answer is no, unless the unlabeled method happens to capture the relevant features and does not need to work for different sets of labels. Due to the lack of existing benchmarks that capture the theoretically expected failure, we introduce a novel type of OOD task, Adjacent OOD detection. This task addresses the critical safety gap caused by significant overlap of ID and OOD data. We show that the Adjacent OOD task accurately captures the failure in unlabeled OOD detection that is hypothesized by our theory. We hope our work will help support more robust research into OOD detection and improve the safety of AI applications.

### ACKNOWLEDGMENTS

This material is based upon work supported by the United States National Science Foundation under grant #2225354.

The authors acknowledge Research Computing at the Rochester Institute of Technology for providing computational resources and support that have contributed to the research results reported in this publication (RIT, 2024).

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

# Appendix

## A   IMPACT STATEMENT

This paper presents work whose goal is to advance the field of machine learning. In particular, this paper seeks to improve the theoretical understanding of safety in machine learning based systems. We hope to positively impact society by improving the work of future researchers and practitioners in building safer AI systems.

## B   PROPERTIES OF MUTUAL INFORMATION AND ENTROPY

In this section we enumerate some of the properties of mutual information that are used to prove the theorems reported in this work. For any random variables $\mathbf{w}, \mathbf{x}, \mathbf{y}$ and $\mathbf{z}$ :

$(P_1)$ Positivity:
$$I(\mathbf{x}; \mathbf{y}) \geq 0, I(\mathbf{x}; \mathbf{y} \mid \mathbf{z}) \geq 0$$

$(P_2)$ Chain rule:
$$I(\mathbf{xy}; \mathbf{z}) = I(\mathbf{y}; \mathbf{z}) + I(\mathbf{x}; \mathbf{z} \mid \mathbf{y})$$

$(P_3)$ Chain rule (Multivariate Mutual Information):
$$I(\mathbf{x}; \mathbf{y}; \mathbf{z}) = I(\mathbf{y}; \mathbf{z}) - I(\mathbf{y}; \mathbf{z} \mid \mathbf{x})$$

$(P_4)$ Positivity of discrete entropy: For discrete $\mathbf{x}$
$$H(\mathbf{x}) \geq 0, H(\mathbf{x} \mid \mathbf{y}) \geq 0$$

$(P_5)$ Entropy and Mutual Information
$$H(\mathbf{x}) = H(\mathbf{x} \mid \mathbf{y}) + I(\mathbf{x}; \mathbf{y})$$

$(P_6)$ Conditioning a variable cannot increase its entropy

$$H(\mathbf{y}|\mathbf{z}) \leq H(\mathbf{y})$$

$(P_7)$ A variable knows about itself as much as any other variable can

$$I(\mathbf{x}; \mathbf{x}) \geq I(\mathbf{x}; \mathbf{y})$$

$(P_8)$ Symmetry of Mutual Information

$$I(\mathbf{x}; \mathbf{y}) = I(\mathbf{y}; \mathbf{x})$$

$(P_9)$ Entropy and Conditional Mutual Information (This is simply $P_5$ conditioned on $\mathbf{z}$)

$$I(\mathbf{x}; \mathbf{y}|\mathbf{z}) = H(\mathbf{x}|\mathbf{z}) - H(\mathbf{x}|\mathbf{yz})$$

$(P_{10})$ Functions of Independent Variables Remain Independent

$$I(\mathbf{x}; \mathbf{y}) = 0 \rightarrow I(f(\mathbf{x}); \mathbf{y}) = 0$$

## C   THEOREMS AND PROOFS OF PREVIOUS WORK

This section contains the supporting theorems and proofs provided by previous work (Federici et al., 2020).

When random variable $\mathbf{z}$ is defined to be a representation of another random variable $\mathbf{x}$, we state that $\mathbf{z}$ is conditionally independent from any other variable in the system once $\mathbf{x}$ is observed. This does not imply that $\mathbf{z}$ must be a deterministic function of $\mathbf{x}$, but that the source of stochasticity for $\mathbf{z}$ is independent of the other random variables. As a result whenever $\mathbf{z}$ is a representation of $\mathbf{x}$ :

$$I(\mathbf{z}; \mathbf{a} \mid \mathbf{xb}) = 0,$$

for any variable (or groups of variables) $\mathbf{a}$ and $\mathbf{b}$ in the system. This condition is accounts for the randomness experienced in training neural networks and the error expected from human labelers. This condition applies to this and the following sections.

### C.1   SUFFICIENCY

**Proposition C.1.** *Let $\mathbf{x}$ and $\mathbf{y}$ be random variables from joint distribution $p(\mathbf{x}, \mathbf{y})$. Let $\mathbf{z}$ be a representation of $\mathbf{x}$, then $\mathbf{z}$ is sufficient for $\mathbf{y}$ if and only if $I(\mathbf{x}; \mathbf{y}) = I(\mathbf{y}; \mathbf{z})$*

*Hypothesis:*

$(H_1)$ $\mathbf{z}$ *is a representation of* $\mathbf{x}$ : $I(\mathbf{y}; \mathbf{z} \mid \mathbf{x}) = 0$

*Thesis:*

$(T_1)$ $I(\mathbf{x}; \mathbf{y} \mid \mathbf{z}) = 0 \iff I(\mathbf{x}; \mathbf{y}) = I(\mathbf{y}; \mathbf{z})$

*Proof.*

$$I(\mathbf{x}; \mathbf{y} \mid \mathbf{z}) \overset{(P_3)}{=} I(\mathbf{x}; \mathbf{y}) - I(\mathbf{x}; \mathbf{y}; \mathbf{z}) \overset{(P_3)}{=} I(\mathbf{x}; \mathbf{y}) - I(\mathbf{y}; \mathbf{z}) + I(\mathbf{y}; \mathbf{z} \mid \mathbf{x})$$
$$\overset{(H_1)}{=} I(\mathbf{x}; \mathbf{y}) - I(\mathbf{y}; \mathbf{z})$$

Since both $I(\mathbf{x}; \mathbf{y})$ and $I(\mathbf{y}; \mathbf{z})$ are non-negative $(P_1)$, $I(\mathbf{x}; \mathbf{y} \mid \mathbf{z}) = 0 \iff I(\mathbf{y}; \mathbf{z}) = I(\mathbf{x}; \mathbf{y})$

$\square$

## D   MAIN THEOREMS AND PROOFS

We ignore cases where the determined variable has an entropy of 0. Generally, if $H(\mathbf{y}|\mathbf{x}) = 0 \to H(\mathbf{y}) > 0$. Also, we only consider cases where the random variables have more than zero entropy.

Note that $R_{\mathbf{x}}$ represents the support of random variable $\mathbf{x}$ such that $R_{\mathbf{x}} = \{\boldsymbol{x} \in \mathbb{R} : P(\boldsymbol{x}) > 0\}$.

### D.1   LOWER BOUND OF MUTUAL INFORMATION FOR SUFFICIENCY

**Lemma D.1.** *Let $\mathbf{x}$ and $\mathbf{y}$ be random variables with joint distribution $p(\mathbf{x}, \mathbf{y})$. Let $\mathbf{z}$ be a representation of $\mathbf{x}$ that is sufficient, as per definition 2.1. Then $I(\mathbf{x}; \mathbf{z}) \geq I(\mathbf{z}; \mathbf{y})$ and $I(\mathbf{x}; \mathbf{z}) \geq I(\mathbf{x}; \mathbf{y})$.*

*Hypothesis:*

$(H_1)$ $\mathbf{z}$ *is a representation of* $\mathbf{x}$ : $I(\mathbf{y}; \mathbf{z} \mid \mathbf{x}) = 0$

$(H_2)$ $\mathbf{z}$ *is a sufficient representation of* $\mathbf{x}$ : $I(\mathbf{x}; \mathbf{y}|\mathbf{z})) = 0$

*Thesis:*

$(T_1)$ $\forall \mathbf{z}.I(\mathbf{x}; \mathbf{z}) \geq I(\mathbf{z}; \mathbf{y}), I(\mathbf{x}; \mathbf{z}) \geq I(\mathbf{x}; \mathbf{y})$

*Proof.* By Construction

$$I(\mathbf{x}\mathbf{y}|\mathbf{z})) \overset{(H_2)}{=} 0$$

$$\overset{(P_2)}{=} I(\mathbf{z}\mathbf{y}; \mathbf{x}) - I(\mathbf{z}; \mathbf{x})$$

$$\overset{(P_2)}{=} I(\mathbf{x}; \mathbf{y}) + I(\mathbf{x}; \mathbf{z}|\mathbf{y}) - I(\mathbf{z}; \mathbf{x})$$

$$\overset{(PropB1)}{=} I(\mathbf{z}; \mathbf{y}) + I(\mathbf{x}; \mathbf{z}|\mathbf{y}) - I(\mathbf{z}; \mathbf{x})$$

$$I(\mathbf{z}; \mathbf{x}) = I(\mathbf{z}; \mathbf{y}) + I(\mathbf{x}; \mathbf{z}|\mathbf{y})$$

$$I(\mathbf{z}; \mathbf{x}) \overset{(P_1)}{\geq} I(\mathbf{z}; \mathbf{y})$$

Note that $I(\mathbf{z}; \mathbf{y}) = I(\mathbf{x}; \mathbf{y})$ for all sufficient representations, as per proposition C.1.

This supports our intuition that the information in the representation consists of relevant information $I(\mathbf{z}; \mathbf{y})$ and irrelevant information $I(\mathbf{x}; \mathbf{z}|\mathbf{y})$. By definition of sufficiency, there must be enough information for $I(\mathbf{z}; \mathbf{y})$ in $I(\mathbf{x}; \mathbf{z})$, which is to say that the size of the encoding cannot be smaller than the minimum size to encode all of $I(\mathbf{x}; \mathbf{y})$.

$\square$

## D.2 CONDITIONAL MUTUAL INFORMATION OF NOISE

**Lemma D.2.** *Let* $\mathbf{x}$ *and* $\mathbf{y}$ *be independent random variables and* $\mathbf{z}$ *be a function of* $\mathbf{x}$ *with joint distribution* $p(\mathbf{x}, \mathbf{y}, \mathbf{z})$. *The conditional mutual information* $I(\mathbf{x}; \mathbf{z}|\mathbf{y})$ *is always equal to the mutual information* $I(\mathbf{x}; \mathbf{z})$. *As in the information content is unchanged when adding noise.*

*Hypothesis:*

$(H_1)$ *Independence of* $\mathbf{x}$ *and* $\mathbf{y}$ *:* $I(\mathbf{x}; \mathbf{y}) = 0$

$(H_2)$ $\mathbf{z}$ *is fully determined by* $\mathbf{x}$ *:* $H(\mathbf{z}|\mathbf{x}) = 0$

*Thesis:*

$(T_1)$ $I(\mathbf{x}; \mathbf{z}|\mathbf{y}) = I(\mathbf{x}; \mathbf{z})$

*Proof.* By Construction.

$(C_1)$ Demonstrates that $H(\mathbf{z}|\mathbf{x}\mathbf{y}) = 0$

$$0 \overset{(P_4)}{\leq} H(\mathbf{z}|\mathbf{x}\mathbf{y}) \overset{(P_6)}{\leq} H(\mathbf{z}|\mathbf{x})$$

$$H(\mathbf{z}|\mathbf{x}\mathbf{y}) \overset{(H_2)}{\leq} 0$$

$(C_2)$ Demonstrates that $I(\mathbf{z}; \mathbf{y}) = 0$

$$I(\mathbf{z}; \mathbf{y}) \overset{(H_2)}{=} I(f(\mathbf{x}); \mathbf{y})$$

$$\overset{(P_{10})}{=} I(\mathbf{x}; \mathbf{y})$$

$$\overset{(H_1)}{=} 0$$

Thus

$$I(\mathbf{x}; \mathbf{z}|\mathbf{y}) \stackrel{(P_9)}{=} H(\mathbf{z}|\mathbf{y}) - H(\mathbf{z}|\mathbf{x}\mathbf{y})$$
$$\stackrel{(C_1)}{=} H(\mathbf{z}|\mathbf{y}) - 0$$
$$\stackrel{(P_5)}{=} H(\mathbf{z}) - I(\mathbf{z}; \mathbf{y})$$
$$\stackrel{(C_2)}{=} H(\mathbf{z}) - 0$$
$$\stackrel{(H_2)}{=} H(\mathbf{z}) - H(\mathbf{z}|\mathbf{x})$$
$$\stackrel{(P_5)}{=} I(\mathbf{x}; \mathbf{z})$$

This supports the intuition that if one added a random noise channel it will not change the mutual information.

$\square$

### D.3 FACTORIZATION OF BOTTLENECK LOSS

**Lemma D.3.** *Let $\mathbf{x}$ be a random variable with label $\mathbf{y}$ such that $H(\mathbf{y}|\mathbf{x}) = 0$ and $\mathbf{z}$ is a sufficient representation of $\mathbf{x}$ for $\mathbf{y}$. The loss function $\mathcal{L} = I(\mathbf{x}; \mathbf{z}) - \beta I(\mathbf{z}; \mathbf{y})$ is equivalent to $\mathcal{L} = H(\mathbf{z}) - \beta I(\mathbf{z}; \mathbf{y})$, with $\beta$ as some constant.*

*Hypothesis:*

$(H_1)$ $\mathbf{z}$ *is fully determined by* $\mathbf{x}$ *:* $H(\mathbf{z}|\mathbf{x}) = 0$

*Thesis:*

$(T_1)$ $I(\mathbf{x}; \mathbf{z}) - \beta I(\mathbf{z}; \mathbf{y}) = H(\mathbf{z}) - \beta I(\mathbf{z}; \mathbf{y})$

*Proof.* By Construction.

$$I(\mathbf{x}; \mathbf{z}) - \beta I(\mathbf{z}; \mathbf{y}) \stackrel{(P_5)}{=} H(\mathbf{z}) - H(\mathbf{z}|\mathbf{x}) - \beta I(\mathbf{z}; \mathbf{y})$$
$$\stackrel{(H_1)}{=} H(\mathbf{z}) - \beta I(\mathbf{z}; \mathbf{y})$$

Due to the relationship between $\mathbf{x}$ and $\mathbf{z}$, we can create an intuitive factorization of the bottleneck loss function. Effectively, we want to maximize $I(\mathbf{z}; \mathbf{y})$ while minimizing the information content of $\mathbf{z}$

$\square$

### D.4 STRICT LABEL BLINDNESS IN THE MINIMAL SUFFICIENT STATISTIC

**Theorem D.4.** *Let $\mathbf{x}$ come from a distribution. $\mathbf{x}$ is composed of two independent variables $\mathbf{x}_1$ and $\mathbf{x}_2$. Let $\mathbf{y}_1$ be a surrogate task such that $H(\mathbf{y}_1|\mathbf{x}_1) = 0$. Let $\mathbf{z}$ be any sufficient representation of $\mathbf{x}$ for $\mathbf{y}_1$ that satisfies the sufficiency definition 2.1 and minimizes the loss function $\mathcal{L} = I(\mathbf{x}_1\mathbf{x}_2; \mathbf{z}) - \beta I(\mathbf{z}; \mathbf{y}_1)$. The possible $\mathbf{z}$ that minimizes $\mathcal{L}$ and is sufficient must meet the condition $I(\mathbf{x}_2; \mathbf{z}) = 0$.*

**Summary** *This proof uses the derivative of the loss function to establish the possible set of local minima that satisfies $\mathcal{L}$. For any possible minima of $\mathcal{L}$, the representation $\mathbf{z}$ must contain information of only $\mathbf{x}_1 -> H(\mathbf{z}|\mathbf{x}_1) = 0$ or only $\mathbf{x}_2 -> H(\mathbf{z}|\mathbf{x}_2) = 0$) or both $\mathbf{x}_1, \mathbf{x}_2 -> H(\mathbf{z}|\mathbf{x}_1, \mathbf{x}_2) = 0$. We show that possible set of all local minima must satisfy $H(\mathbf{z}|\mathbf{x}_1) = 0$ by showing that the other two cases must always have greater $\mathcal{L}$. This proves the Theorem that the learned representation cannot contain information about $\mathbf{x}_2$.*

$(H_1)$ $\mathbf{z}$ *is fully determined by* $\mathbf{x}$ *:* $H(\mathbf{z}|\mathbf{x}) = 0$

$(H_2)$ **z** *is a representation of* **x** $: I(\mathbf{y}; \mathbf{z} \mid \mathbf{x}) = 0$

$(H_3)$ **z** *is a sufficient representation of* **x** $: I(\mathbf{x}; \mathbf{y}|\mathbf{z}) = 0$

$(H_4)$ **x** *is composed of two independent variables* $\mathbf{x}_1, \mathbf{x}_2 : \mathbf{x} = \mathbf{x}_1, \mathbf{x}_2, I(\mathbf{x}_1; \mathbf{x}_2) = 0$

$(H_5)$ **y** *is fully determined by* $\mathbf{x}_1$*:* $H(\mathbf{y}|\mathbf{x}_1) = 0$

*Thesis:*

$(T_1)\ \forall \mathbf{z}.I(\mathbf{x}_2, \mathbf{z}) = 0$

*Proof.* By Construction

$(C_1)$ demonstrates that $\mathcal{L} = H(\mathbf{z}) - \beta I(\mathbf{z}; \mathbf{y})$ via factoring $I(\mathbf{x}_1\mathbf{x}_2; \mathbf{z})$. Alternatively, Theorem D.3 creates the same result.

$$
\begin{aligned}
I(\mathbf{x}_1, \mathbf{x}_2; \mathbf{z}) &\overset{(P_2)}{=} I(\mathbf{x}_1; \mathbf{z}) + I(\mathbf{x}_2; \mathbf{z}|\mathbf{x}_1) \\
&\overset{(P_5)}{=} H(\mathbf{z}) - H(\mathbf{z}|\mathbf{x}_1) + I(\mathbf{x}_2; \mathbf{z}|\mathbf{x}_1) \\
&\overset{(P_9)}{=} H(\mathbf{z}) - H(\mathbf{z}|\mathbf{x}_1) + H(\mathbf{z}|\mathbf{x}_1) - H(\mathbf{z}|\mathbf{x}_1\mathbf{x}_2) \\
&\overset{(H_1)}{=} H(\mathbf{z}) - H(\mathbf{z}|\mathbf{x}_1) + H(\mathbf{z}|\mathbf{x}_1) - 0 \\
\mathcal{L} &= H(\mathbf{z}) - \beta I(\mathbf{z}; \mathbf{y})
\end{aligned}
$$

$(C_2)$ Demonstrates that $I(\mathbf{z}; \mathbf{y}) = I(\mathbf{x}; \mathbf{y})$ as per Theorem C.1.

$(C_3)$ Demonstrates that $I(\mathbf{z}; \mathbf{y})$ is a constant across all sufficient representations because Theorem C.1 applies.

$(C_4)$ Demonstrates that for all possible **z** satisfying $(H_3)$, their loss can be compared using only $\mathcal{L}_z = H(\mathbf{z})$ for comparing across **z**

$$
\begin{aligned}
\frac{d\mathcal{L}}{d\mathbf{z}} &\overset{(C_1)}{=} \frac{H(\mathbf{z})}{d\mathbf{z}} - \frac{\beta I(\mathbf{z}; \mathbf{y})}{d\mathbf{z}} \\
&\overset{(C_3)}{=} \frac{H(\mathbf{z})}{d\mathbf{z}} - 0
\end{aligned}
$$

$(C_5)$ Demonstrates that the value of $H(\mathbf{z})$ at all possible **z** that minimizes $\mathcal{L}$ is the same. Even for different minimal **z**, they must have the same $H(\mathbf{z})$ to all be minimal. When comparing possible minimal solutions to $\mathcal{L}$, $H(\mathbf{z})$ is constant across all minimal solutions.

$(C_6)$ Demonstrates that any **z** that satisfies sufficiency must satisfy $I(\mathbf{z}; \mathbf{x}) \geq I(\mathbf{z}; \mathbf{y})$ and $I(\mathbf{z}; \mathbf{x}) \geq I(\mathbf{x}; \mathbf{y})$ as per Theorem D.1.

$(C_7)$ Demonstrates that minima(s) exists only where $H(\mathbf{z}) = I(\mathbf{z}; \mathbf{y})$ and $H(\mathbf{z}|\mathbf{x}) = 0$. Note that $H(\mathbf{z}) = I(\mathbf{x}; \mathbf{y}) = I(\mathbf{z}; \mathbf{y})$ is the most compact representation size that is sufficient.

$$
\begin{aligned}
I(\mathbf{z}; \mathbf{x}) &\overset{(C_6)}{\geq} I(\mathbf{z}; \mathbf{y}) \\
H(\mathbf{z}) - H(\mathbf{z}|\mathbf{x}) &\overset{(P_5)}{\geq} I(\mathbf{z}; \mathbf{y}) \\
\forall \mathbf{z}|C_6 \wedge H_3 \wedge I(\mathbf{z}; \mathbf{x}) > I(\mathbf{z}; \mathbf{y}).\exists \mathbf{z}'|\mathbf{z}' &= f(\mathbf{z}) \wedge I(\mathbf{z}; \mathbf{x}) > I(\mathbf{z}'; \mathbf{x}) \wedge C_6 \wedge H_3
\end{aligned}
$$

From $(C_7)$ there exists only 3 types of minimas, separated by their dependence on the variables $\mathbf{x}_1, \mathbf{x}_2$. As per $(H_1)$, any **z** must follow one of the 3 types.

1. Dependent only on $\mathbf{x}_1$: $\forall \mathbf{z}|H(\mathbf{z}|\mathbf{x}_1) = 0 \rightarrow I(\mathbf{x}_2; \mathbf{z}) = 0$

2. Dependent only on $\mathbf{x}_2$: $\forall \mathbf{z}|H(\mathbf{z}|\mathbf{x}_2) = 0 \rightarrow I(\mathbf{x}_2; \mathbf{z}) > 0$

3. Dependent on both $\mathbf{x}_1\mathbf{x}_2$: $\forall \mathbf{z} | H(\mathbf{z}|\mathbf{x}_1, \mathbf{x}_2) = 0 \land H(\mathbf{z}|\mathbf{x}_1) > 0 \land H(\mathbf{z}|\mathbf{x}_2) > 0 \rightarrow I(\mathbf{x}_2; \mathbf{z}) > 0$

From here we will show that all type 2 and type 3 minimas always fail $(H_3)$ or have greater $\mathcal{L}$ than any type 1 minima.

**Type 1** $\mathbf{x}_1$: $\forall \mathbf{z} | H(\mathbf{z}|\mathbf{x}_1) = 0 \rightarrow I(\mathbf{x}_2; \mathbf{z}) = 0$

$(C_8)$ Demonstrates that there exists $H(\mathbf{z}) = I(\mathbf{z}; \mathbf{y}) = I(\mathbf{x}_1; \mathbf{z})$ and it is a set of minimas satisfying $(C_7)$. This also establishes an upper bound for solutions to $\mathcal{L}$ due to $(C_5)$. Therefore, any solution for type 1, type 2, and type 3 must satisfy $I(\mathbf{z}; \mathbf{y}) \leq I(\mathbf{x}_1; \mathbf{z})$ to be sufficient and $I(\mathbf{z}; \mathbf{y}) = I(\mathbf{x}_1; \mathbf{z})$ to be minimal.

$$
\begin{aligned}
I(\mathbf{z}; \mathbf{y}) &\overset{(C_6)}{\leq} I(\mathbf{z}; \mathbf{x}) \\
&\overset{(H_4)}{\leq} I(\mathbf{x}_1, \mathbf{x}_2; \mathbf{z}) \\
&\overset{(P_2)}{\leq} I(\mathbf{x}_2; \mathbf{z}) + I(\mathbf{x}_1; \mathbf{z}|\mathbf{x}_2) \\
&\overset{(Type1)}{\leq} 0 + I(\mathbf{x}_1; \mathbf{z}|\mathbf{x}_2) \\
&\overset{(TheorumD.2)}{\leq} I(\mathbf{x}_1; \mathbf{z}) \\
&\overset{(P_5)}{\leq} H(\mathbf{z}) - H(\mathbf{z}|\mathbf{x}_1) \\
\exists \mathbf{z} | I(\mathbf{x}_1; \mathbf{z}) &= I(\mathbf{z}; \mathbf{y}) = I(\mathbf{x}; \mathbf{z}) = I(\mathbf{x}; \mathbf{y})
\end{aligned}
$$

$(C_9)$ Demonstrates that there exists no $H(\mathbf{z}') < H(\mathbf{z})$ that satisfies sufficiency if $\mathbf{z}$ satisfies $(C_8)$ and is also $I(\mathbf{z}; \mathbf{x}_2) = 0$.

$$
\begin{aligned}
C_8 &\rightarrow I(\mathbf{x}_1; \mathbf{z}) = I(\mathbf{x}; \mathbf{y}) \\
H(\mathbf{z}') < H(\mathbf{z}) &\rightarrow I(\mathbf{x}_1; \mathbf{z}') < I(\mathbf{x}_1; \mathbf{z}) \\
&\rightarrow \neg(C_2) : I(\mathbf{x}_1; \mathbf{z}') < I(\mathbf{x}_1; \mathbf{z}) = I(\mathbf{y}; \mathbf{z}) = I(\mathbf{x}; \mathbf{y})
\end{aligned}
$$

**Type 2** $\mathbf{x}_2$: $\forall \mathbf{z} | H(\mathbf{z}|\mathbf{x}_2) = 0 \rightarrow I(\mathbf{x}_2; \mathbf{z}) > 0$

$(C_{10})$ Demonstrates that no type 2 minima can exist, simply because it would contain no information regarding $\mathbf{x}_1$, thus failing to satisfy $(H_3)$. This is because $\mathbf{z}$ cannot contain any information about $\mathbf{x}_1$, otherwise we would not satisfy $H(\mathbf{z}|\mathbf{x}_2) = 0$. If the representation $\mathbf{z}$ contains no information about $\mathbf{y}$, then it is not sufficient.

$$
\begin{aligned}
H(\mathbf{z}|\mathbf{x}_2) = 0 &\rightarrow \mathbf{z} = f(\mathbf{x}_2) \\
0 &\overset{(H_4)}{=} I(\mathbf{x}_1; \mathbf{x}_2) \\
&\overset{(P_{10})}{=} I(f(\mathbf{x}_1); \mathbf{x}_2) \\
&\overset{(H_5)}{=} I(\mathbf{y}; \mathbf{x}_2) \\
&\overset{(P_{10})}{=} I(\mathbf{y}; f(\mathbf{x}_2)) \\
0 &= I(\mathbf{y}; \mathbf{z})
\end{aligned}
$$

**Type 3** $\mathbf{x}_1, \mathbf{x}_2$: $\forall \mathbf{z} | H(\mathbf{z}|\mathbf{x}_1, \mathbf{x}_2) = 0 \land H(\mathbf{z}|\mathbf{x}_1) > 0 \land H(\mathbf{z}|\mathbf{x}_2) > 0 \rightarrow I(\mathbf{x}_2; \mathbf{z}) > 0$

$(C_{11})$ Demonstrates that any $\mathbf{z}$ that could be minimal must also satisfy $(C_8)$ for sufficiency. Note that $(C_8)$ implies that any $I(\mathbf{x}_1; \mathbf{z}) > I(\mathbf{z}; \mathbf{y})$ is not minimal.

$$I(\mathbf{z};\mathbf{y}) \overset{(C_6)}{\leq} I(\mathbf{z};\mathbf{x})$$
$$\overset{(H_4)}{\leq} I(\mathbf{x}_1\mathbf{x}_2;\mathbf{z})$$
$$I(\mathbf{z};\mathbf{y}) \overset{(P_2)}{\leq} I(\mathbf{x}_1;\mathbf{z}) + I(\mathbf{x}_2;\mathbf{z}|\mathbf{x}_1)$$
$$(C_8) \to I(\mathbf{x}_1;\mathbf{z}) = I(\mathbf{z};\mathbf{y})$$

$(C_{12})$ Demonstrates that any $\mathbf{z}'$ where $I(\mathbf{z}';\mathbf{x}_2) > I(\mathbf{z};\mathbf{x}_2)$ and $I(\mathbf{z};\mathbf{x}_2) = 0$ that maintains $H(\mathbf{z}') = H(\mathbf{z})$ results in solutions that are not sufficient as required by $(H_3)$ because we know that the size of the representation must be at least $I(\mathbf{x};\mathbf{y})$ as defined in $(C_6)$

$$C_8 \to H(\mathbf{z}) \text{ is constant across all minima}$$
$$C_8 \to H(\mathbf{z}) = H(\mathbf{z}') \text{ for } \mathbf{z}' \text{ to be minimal}$$
$$C_8 \to I(\mathbf{x}_1;\mathbf{z}) = I(\mathbf{x};\mathbf{y})$$
$$I(\mathbf{x}_2;\mathbf{z}) = 0 \to H(\mathbf{z}|\mathbf{x}_1) = 0$$
$$\forall \mathbf{z}'|I(\mathbf{x}_2;\mathbf{z}') > 0 : H(\mathbf{z}'|\mathbf{x}_1) > H(\mathbf{z}|\mathbf{x}_1)$$
$$H(\mathbf{z}'|\mathbf{x}_1) > H(\mathbf{z}|\mathbf{x}_1) \to H(\mathbf{z}') - H(\mathbf{z}'|\mathbf{x}_1) < H(\mathbf{z}) - H(\mathbf{z}|\mathbf{x}_1)$$
$$\overset{(P_5)}{\to} I(\mathbf{x}_1;\mathbf{z}') < I(\mathbf{x}_1;\mathbf{z})$$
$$\to \neg(C_6) : I(\mathbf{x}_1;\mathbf{z}') < I(\mathbf{x};\mathbf{y})$$

$(C_{13})$ Demonstrates that combining $(C_{11})$ and $(C_{12})$, there is no type 3 solution that has an equal $\mathcal{L}$ to the minimal type 1 solution that also maintains sufficiency $(H_3)$ and $(C_6)$. This confirms the definition of entropy, in that encoding more independent information requires more bits or nats.

This means that only a type 1 solution can be both minimal and sufficient, which proves the thesis.

To summarize this proof, we can compare the losses of all sufficient solutions with $\mathcal{L} = H(\mathbf{z})$. Of those sufficient solutions, the one that minimizes $\mathcal{L}$ is the one with the smallest $H(\mathbf{z})$. The minimal sufficient representation is $\mathbf{z}$ that captures only all of $I(\mathbf{x}_1;\mathbf{y})$ and nothing else. Thus the minimal $\mathbf{z}$ cannot have $I(\mathbf{x}_2;\mathbf{z}) > 0$ because such $\mathbf{z}$ would encode information outside of $I(\mathbf{x}_1;\mathbf{y})$.

$\square$

## D.5 Independence of Filtered Distributions

**Lemma D.5.** *Let* $\mathbf{x}$ *come from a distribution.* $\mathbf{x}$ *is composed of two independent variables* $\mathbf{x}_1$ *and* $\mathbf{x}_2$. *For* $\mathbf{x}_2'$ *where* $R_{\boldsymbol{x}_2'} \subset R_{\boldsymbol{x}_2}$, *there exists no* $\mathbf{x}_2'$ *such that* $H(\mathbf{x}_1|\mathbf{x}_2') < H(\mathbf{x}_1)$.

**Summary** *This proof uses the chain rule of mutual information to show that contradiction arises if* $\mathbf{x}_2'$ *could filter* $\mathbf{x}_1$ *in a non random way.*

*Hypothesis:*

$(H_1)$ $\mathbf{x}_2'$ *is fully determined by* $\mathbf{x}_2$ : $H(\mathbf{x}_2'|\mathbf{x}_2) = 0$ *where* $R_{\boldsymbol{x}_2'} \subset R_{\boldsymbol{x}_2}$

$(H_2)$ *Independence of* $\mathbf{x}_1$ *and* $\mathbf{y}_2$ : $I(\mathbf{x}_1;\mathbf{x}_2) = 0$

*Thesis:*

$(T_1)$ $\nexists \mathbf{x}_2'.H(\mathbf{x}_1|\mathbf{x}_2) < H(\mathbf{x}_1)$

*Proof.* by contradiction $H(\mathbf{x}_1|\mathbf{x}_2) < H(\mathbf{x}_1)$

$(C_1)$ Demonstrates $I(\mathbf{x}_2;\mathbf{x}_2') = I(\mathbf{x}_2';\mathbf{x}_2')$

$$I(\mathbf{x}_2; \mathbf{x}_2') \stackrel{(P_4)}{=} H(\mathbf{x}_2') - H(\mathbf{x}_2'|\mathbf{x}_2)$$
$$\stackrel{(H_1)}{=} H(\mathbf{x}_2') - 0$$
$$\stackrel{(P_3)}{=} H(\mathbf{x}_2') - H(\mathbf{x}_2'|\mathbf{x}_2')$$
$$\stackrel{(P_3)}{=} I(\mathbf{x}_2'; \mathbf{x}_2')$$

$(C_2)$ Demonstrates $I(\mathbf{x}_2'; \mathbf{x}_1|\mathbf{x}_2) = 0$

$$I(\mathbf{x}_2'; \mathbf{x}_1|\mathbf{x}_2) \stackrel{(P_2)}{=} I(\mathbf{x}_2'; \mathbf{x}_2\mathbf{x}_1) - I(\mathbf{x}_2; \mathbf{x}_2')$$
$$\stackrel{(C_1)}{=} I(\mathbf{x}_2'; \mathbf{x}_2\mathbf{x}_1) - I(\mathbf{x}_2'; \mathbf{x}_2')$$
$$I(\mathbf{x}_2'; \mathbf{x}_1|\mathbf{x}_2) \stackrel{(P_7)}{\leq} 0 \leftarrow I(\mathbf{x}_2'; \mathbf{x}_2\mathbf{x}_1) \leq I(\mathbf{x}_2'; \mathbf{x}_2')$$
$$\stackrel{(P_1)}{\geq} 0$$
$$= 0$$

$(C_3)$ Demonstrates $I(\mathbf{x}_1; \mathbf{x}_2') > 0$ via non independence implied by $\neg T_1$

Contradiction arises when we consider symmetric applications of the chain rule to $I(\mathbf{x}_1; \mathbf{x}_2\mathbf{x}_2')$

$$I(\mathbf{x}_1; \mathbf{x}_2'\mathbf{x}_2) \stackrel{(P_2)}{=} I(\mathbf{x}_1; \mathbf{x}_2') + I(\mathbf{x}_1; \mathbf{x}_2|\mathbf{x}_2')$$
$$I(\mathbf{x}_1; \mathbf{x}_2'\mathbf{x}_2) \stackrel{(C_3)}{>} 0$$
$$I(\mathbf{x}_1; \mathbf{x}_2\mathbf{x}_2') \stackrel{(P_2)}{=} I(\mathbf{x}_1; \mathbf{x}_2) + I(\mathbf{x}_1; \mathbf{x}_2'|\mathbf{x}_2)$$
$$\stackrel{(C_2)}{=} I(\mathbf{x}_1; \mathbf{x}_2)$$
$$\stackrel{(H_2)}{=} 0$$

Since $I(\mathbf{x}_1; \mathbf{x}_2\mathbf{x}_2')$ cannot be both zero and greater than zero, $\neg T_1$ creates a contradiction, which supports $T_1$.

It is easy to confuse this with the existence of a non independent subset $\mathbf{C} := \mathbf{A} \cap \mathbf{B}$, where $\mathbf{A}, \mathbf{B}$ are independent events. However, this example violates $(H_1)$, since we cannot determine $\mathbf{C}$ using only $\mathbf{A}$ or only $\mathbf{B}$.

$\square$

### D.6 STRICT LABEL BLINDNESS IN FILTERED DISTRIBUTIONS - GUARANTEED OOD FAILURE

**Corollary D.6.** *Let* $\mathbf{x}$ *come from a distribution.* $\mathbf{x}$ *is composed of two independent variables* $\mathbf{x}_1$ *and* $\mathbf{x}_2$. *Let* $\mathbf{y}_1$ *be a a surrogate task such generated by* $\mathbf{y}_1 = f_1(\boldsymbol{x}_1)$ $H(\mathbf{y}_1|\mathbf{x}_1) = 0$. *Let* $\mathbf{y}_2$ *be a label such that* $H(\mathbf{y}_2|\mathbf{x}_2) = 0$ *and* $\mathbf{y}_2 = f_2(\boldsymbol{x}_2)$. *Let* $\mathbb{Y}_{in}$ *be as subset of labels* $\mathbb{Y}_{in} \subset R_{\mathbf{y}_2}$. *Let* $\mathbf{x}'$ *be a subset of* $\mathbf{x}$ *where* $R_{\mathbf{x}'} = R_{\mathbf{x}} \cap \{\boldsymbol{x} \in \mathbb{R} : f_2(\boldsymbol{x}_2) \in \mathbb{Y}_{in}\}$ *such that* $\mathbf{x}'$ *is composed of independent variables* $\mathbf{x}_1'$ *and* $\mathbf{x}_2'$ *and* $\mathbf{y}_1' = f_1(\boldsymbol{x}_1')$. *The sufficient representation* $\mathbf{z}$ *learned by minimizing* $\mathcal{L} = I(\mathbf{x}_1', \mathbf{x}_2'; \mathbf{z}) - \beta I(\mathbf{z}; \mathbf{y}_1')$ *must have* $I(\mathbf{x}_2; \mathbf{z}) = 0$ *and* $I(\mathbf{y}_2; \mathbf{z}) = 0$.

**Summary** *This proof combines Theorem D.5 and Theorem D.4.*

*Hypothesis:*

$(H_1)$ $\mathbf{z}$ *is fully determined by* $\mathbf{x}$ *:* $H(\mathbf{z}|\mathbf{x}) = 0$

$(H_2)$ **z** *is a representation of* **x** *:* $I(\mathbf{y}; \mathbf{z} \mid \mathbf{x}) = 0$

$(H_3)$ **z** *is a sufficient representation of* **x** *:* $I(\mathbf{x}; \mathbf{y}|\mathbf{z}) = 0$

$(H_4)$ **x** *is composed of two independent variables* $\mathbf{x}_1, \mathbf{x}_2 :$ $\mathbf{x} = \mathbf{x}_1\mathbf{x}_2, I(\mathbf{x}_1; \mathbf{x}_2) = 0$

$(H_5)$ **y** *is fully determined by* $\mathbf{x}_1$*:* $H(\mathbf{y}|\mathbf{x}_1) = 0$

$(H_6)$ $\mathbf{x}'$ *is a subset of* **x** *filtered by* $\mathbb{Y}_{in} :$ $R_{\mathbf{x}'} = R_{\mathbf{x}} \cap \{\boldsymbol{x} \in \mathbb{R} : f_2(\boldsymbol{x}_2) \in \mathbb{Y}_{in}\}$

*Thesis:*

$(T_1)$ $\forall \mathbf{z}.I(\mathbf{x}_2; \mathbf{z}) = 0, I(\mathbf{x}'_2; \mathbf{z}) = 0$

*Proof.* By Construction.

$(C_1)$ Demonstrates that $I(\mathbf{x}'_1; \mathbf{x}'_2) = 0$ due to Lemma D.5

Using $(P_{10})$, we know that independent functions stay independent and thus $I(\mathbf{x}'_1; \mathbf{x}_2) = 0, I(\mathbf{x}'_1; \mathbf{x}_2) = 0$. From TheoremD.4 we know that encoding an variable independent of the target y results in a higher loss, therefore $I(\mathbf{x}'_2; \mathbf{z}) = 0$ and $I(\mathbf{x}_2; \mathbf{z}) = 0$ since both are independent of $\mathbf{x}'_1$.

By combining Lemma D.5 and TheoremD.4, we know that any surrogate learning objective independent of a downstream objective (say classifying labels) results in a representation containing no information for the downstream objective. If it contains no information for one objective, it contains no information for derivitives of that objective (eg. no label information means no OOD detection information).

$\square$

### D.7 UNAVOIDABLE RISK OF OVERLAPPING OUT OF DISTRIBUTION DATA

**Theorem D.7.** *Let* **x** *come from a distribution. Let* $f$ *be some labeling function to generate labels* **y** *such that* $y = f(\boldsymbol{x})$*, where there are at least two unique labels* $|R_{\mathbf{y}}| > 1$*. Let* $\mathbf{x}_{in}$ *be a random subset of* **x** *where* $R_{\mathbf{x}_{in}} \subsetneq R_{\mathbf{x}}$ *and* $|R_{\mathbf{x}_{in}}| < \infty$*. Let* $\mathbf{y}_{in}$ *be labels generated from* $\boldsymbol{y}_{in} = f(\boldsymbol{x}_{in})$*. The probability that a randomly selected* $\boldsymbol{x}$ *contains* $\boldsymbol{y}$ *not present in* $R_{\mathbf{y}_{in}}$ *is always greater than 0.*

*Hypothesis:*

$(H_1)$ **x** *comes from any distribution*

$(H_2)$ **y** *is a label generated from function* $\boldsymbol{y} = f(\boldsymbol{x})$ *such that* $|R_{\mathbf{y}}| > 1$

$(H_3)$ $\mathbf{x}_i$ *is a random subset of* **x** *where* $R_{\mathbf{x}_{in}} \subsetneq R_{\mathbf{x}}$ *and* $|R_{\mathbf{x}_{in}}| < \infty$ *and* $\boldsymbol{y}_{in} = f(\boldsymbol{x}_{in})$*.*

*Thesis*

$(T_1)$ $\forall \mathbf{x}.P(f(\boldsymbol{x}) \notin R_{\mathbf{y}_{in}}) > 0)$

*Proof.* by contradiction $(\neg T_1)$ $P(f(\boldsymbol{x}) \notin R_{\mathbf{y}_{in}}) = 0\}$

$(C_1)$ Demonstrates that $\forall \mathbf{x}_i.R_{\mathbf{y}_{in}} = R_{\mathbf{y}}$ because there must exist no sample $\boldsymbol{x}$ such that $f(\boldsymbol{x}) \notin R_{\mathbf{y}_{in}}$.

$(C_2)$ Demonstrates that $\forall \boldsymbol{y}_n.P(f(\boldsymbol{x}) = \boldsymbol{y}_n) > 0$, where $\boldsymbol{y}_n \in R_{\mathbf{y}}$

Contradiction arises when we consider that it is possible to sample the same label $\boldsymbol{y}_n$ for any finite number of repetitions, as per $(C_2)$. This would create a set of any finite size consisting only of the label $\boldsymbol{y}_n$. Thus, there always exists $R_{\mathbf{y}_{in}} \subsetneq R_{\mathbf{y}}$ which contradicts $(C_1)$.

More realistically, $R_{\mathbf{y}_{in}}$ can consist of all elements of $R_{\mathbf{y}}$ except one and still guarantee $P(f(\boldsymbol{x}) \notin R_{\mathbf{y}_{in}}) > 0)$.

$\square$

# E    SAMPLE IMAGES FROM DATASETS USED IN THE EXPERIMENTS

Space intentionally left blank.

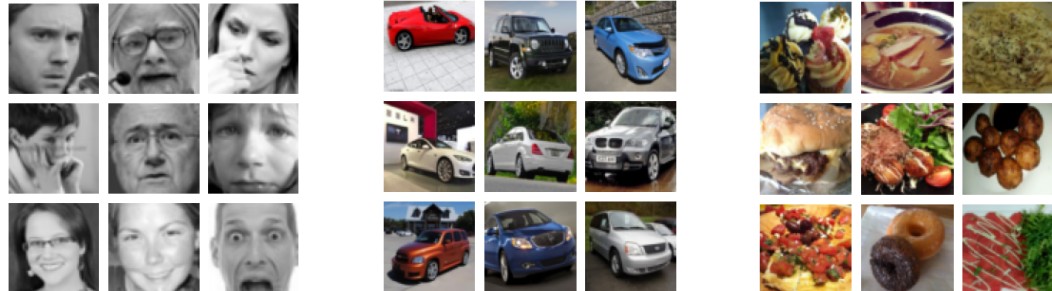

Figure 2: From left to right, sample images from the datasets ICML Facial Expression, Stanford Cars, and Food 101. These datasets contain classes that are visually similar, in contrast to CIFAR10, which includes classes such as airplane and dog that are not visually similar.

The above figure shows sample images from the datasets used in the experiments.

## F    ADDITIONAL EXPERIMENTAL RESULTS

### F.1    ADJACENT OOD ON CIFAR10 AND CIFAR100

See table 2 for results on CIFAR10 and CIFAR100 adjacent OOD benchmarks. SSL methods perform better than on the Faces, Cars, and Food dataset.

|  | CIFAR10 |  | CIFAR100 |  |
|---|---|---|---|---|
| Method | AUROC | FPR95 | AUROC | FPR95 |
| Supervised MSP | 85.3±5.9 | 73.0±9.1 | 78.3±0.9 | 80.9±1.2 |
| SimCLR KNN | 77.6±8.0 | 75.6±6.0 | 68.8±1.7 | 88.8±1.9 |
| SimCLR SSD | 77.6±8.0 | 69.1±1.3 | 70.2±1.2 | 88.2±2.4 |
| RotLoss KNN | 71.4±9.1 | 83.3±8.3 | 48.1±2.2 | 94.2±1.5 |
| RotLoss SSD | 71.1±6.7 | 82.6±9.0 | 47.9±2.3 | 96.1±0.5 |

Table 2: Adjacent OOD Performance on CIFAR10 and CIFAR100.

### F.2    FAR OOD DETECTION PERFORMANCE

A SimCLR KNN based SSL OOD detection method performs extremely well on far OOD tasks. See table 3

| ID Data vs | AUROC | FPR95 |
|---|---|---|
| OOD CIFAR10 |  |  |
| ICML Faces | 99.7±0.1 | 0.1±0.0 |
| Stanford Cars | 98.1±0.1 | 8.1±0.9 |
| Food 101 | 99.8±0.1 | 0.0±0.0 |
| OOD CIFAR100 |  |  |
| ICML Faces | 99.7±0.1 | 0.2±0.1 |
| Stanford Cars | 99.2±0.2 | 1.5±0.1 |
| Food 101 | 99.4±0.1 | 1.4±0.1 |

Table 3: Using CIFAR10 and CIFAR100 as OOD sets, we see that far OOD detection performance for the SimCLR KNN method is very good. Unlabeled OOD detection methods tend to perform very well in far OOD tasks, see (Sehwag et al., 2021; Tack et al., 2020; Liu et al., 2023; Guille-Escuret et al., 2024; Wang et al., 2023)

## G    REVIEWING CLIPN PRETRAINING DATA

We consider sample images from CC3M that contain the label from their respective benchmark. We observe in figure 3 that images containing the word angry often do not contain a human face. This is in contrast to the labels for the Cars and Food dataset, where the pretraining data is very similar to the benchmarking data, see figure 4 and 5.

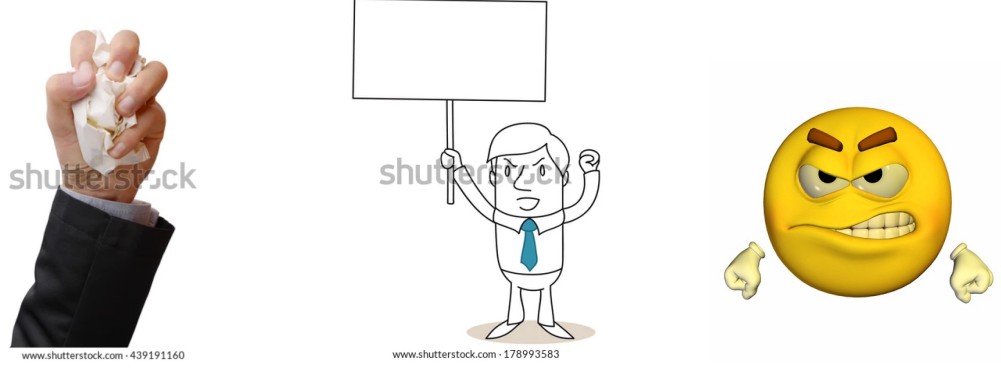

Figure 3: Comparing Images from the CC3M Dataset with captions containing the word angry. These are drastically different from the images in ICML face dataset. Images captioned with other facial expressions also tend to lack a human face.

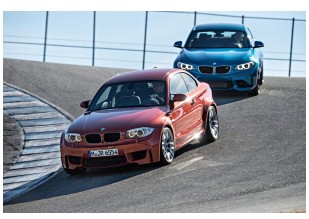 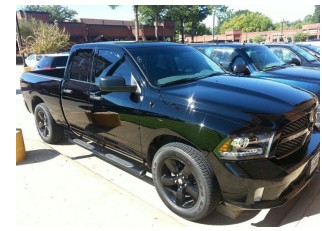 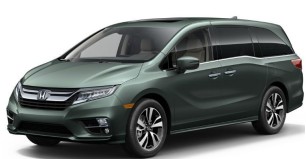

Figure 4: Comparing Images from the CC3M Dataset with captions containing the word BMW 3 Series, Dodge Ram, and Honda Odyssey, left to right. These are are quite similar to the images in Cars dataset

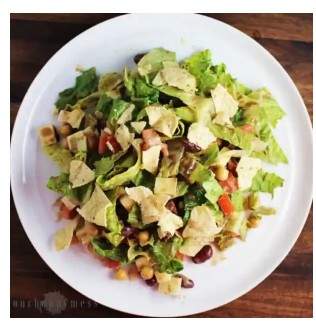 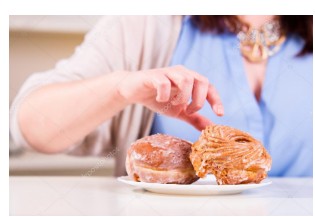 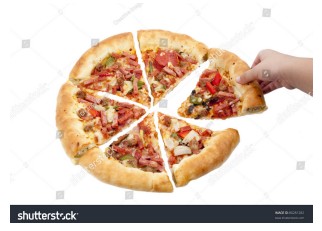

Figure 5: Comparing Images from the CC3M Dataset with captions containing the word Caeser Salad, Donut, and Pizza, left to right. These are are quite similar to the images in Food 101 dataset

