# OpenReview forum: "Can We Ignore Labels in Out of Distribution Detection?"
_ICLR.cc/2025/Conference — ICLR 2025 Poster_

### Official Review · Reviewer_yEDC · 2024-10-18

**Soundness:** 3
**Presentation:** 3
**Contribution:** 3
**Rating:** 6
**Confidence:** 3

**Summary:**

This paper provides a theory to explain the failure of unlabeled OOD detection. The authors also define a new OOD task to test for label blindness and conduct experiments to verify the correctness of their theorem.

**Strengths:**

[1] The paper is clear and easy to understand.

[2] The theory clearly explains the potential problem of unlabeled OOD detection.

[3] The theory is verified by the empirical studies, raising real concerns about unlabeled OOD detection.

**Weaknesses:**

[1] While the main theory is sound in this paper, it is intuitive that the unlabeled OOD methods will fail if there is zero information between the learning objective and the label: For example, if we use linear regression to fit random noise, it is supposed to learn nothing. If the learning objective of the unlabeled OOD detection learns every feature but still has zero information with the label, then my understanding is that labeled OOD methods will also fail. Could the authors provide some comments on how the theory should be adapted in such a case?

[2] While Theorem 4.1 provides some justification on the effectiveness of the proposed Adjacent OOD task, there is a gap between Theorem 4.1 and the main theory: even if P(a randomly selected x contains y not present in $R_{y_{in}}$)>0, this is not exactly connected with the mutual information between the learning objective and the label. I would suggest the authors provide some more evidence, either theory or just simple simulation, to provide a more rigorous connection between the main theory and conclusion of Theorem 4.1.

[3] It is not clear how the theory can provide practical guidance on designing better learning objectives of unlabeled OOD. The paper provides the Adjacent OOD task, but this is different from suggesting a way of improving existing design: we know they fails, but we don't know how to improve them. Could the authors provide some comments on this?

[4] Please highlight the technical contribution when deriving the theorems. Since as elaborated in Weakness [1], the main theory is intuitive, it is not clear how difficult it is to prove it.

[5] Writing: please revise the wording in this paper to be more objective. For example, the word "you" or "your" appear 6 times in total.

**Questions:**

Please address my comments in the weakness section.

---

> ### Author Response · Authors · 2024-11-22
> **Response to Reviewer 3**
>
> Thank you for your thoughtful and constructive feedback. We address your comments and questions below.
>
> ### Q1. While the main theory is sound in this paper, it is intuitive that the unlabeled OOD methods will fail if there is zero information between the learning objective and the label: For example, if we use linear regression to fit random noise, it is supposed to learn nothing. If the learning objective of the unlabeled OOD detection learns every feature but still has zero information with the label, then my understanding is that labeled OOD methods will also fail. Could the authors provide some comments on how the theory should be adapted in such a case?
>
> A1. In the example where linear regression is fit to random noise, the representation would be some function of the random noise. In this case, the representation would contain no information on the label as the random noise contains no information on the label, due to the property of conditional entropy. Labeled OOD methods would also fail in this case. If there is no useful information in the input for predicting the label at all (as implied by every feature containing zero information), then nothing can be learned about the label from the input using any method.
>
> When we consider the case where an unlabeled OOD method learns a representation that contains every feature in the input and also contains zero information about the label, then we must conclude that every feature in the input is not useful for predicting the label at all. This would devolve into the case above, where nothing can be learned about the label from the input using either unlabeled or labeled methods.
>
>
> ### Q2. While Theorem 4.1 provides some justification on the effectiveness of the proposed Adjacent OOD task, there is a gap between Theorem 4.1 and the main theory: even if P(a randomly selected x contains y not present in ..., this is not exactly connected with the mutual information between the learning objective and the label. I would suggest the authors provide some more evidence, either theory or just simple simulation, to provide a more rigorous connection between the main theory and conclusion of Theorem 4.1.
> A2. We have added a paragraph connecting the two concepts by noting how the presence of adjacent data is what causes the label blindness failure. We also note in that paragraph that OOD benchmarks on non adjacent data do not show label blindness.
>
> ### Q3. It is not clear how the theory can provide practical guidance on designing better learning objectives of unlabeled OOD. The paper provides the Adjacent OOD task, but this is different from suggesting a way of improving existing design: we know they fails, but we don't know how to improve them. Could the authors provide some comments on this?
>
> A3. We have updated Section 7 to include more discussion about designing better OOD methods. The authors hope that the theoretical insights of this paper will inform current researchers in OOD detection to improve discussion on the limitations of unlabeled OOD detection. Purely unlabeled methods may be limited to cases where their learning objective is aligned with the labels in question. We hope that researchers exploring completely unlabeled methods would consider these limitations and also explore labeled or minimally labeled methods, as in (Sehwag et al 2021) where they consider their method in both unlabeled and labeled settings.
>
> ### Q4. Please highlight the technical contribution when deriving the theorems. Since as elaborated in Weakness [1], the main theory is intuitive, it is not clear how difficult it is to prove it.
>
> A4. We have added a summary in the appendix for theorem 3.1, as that is the most complex proof in the theoretical work. Shorter summaries for theorems 3.2 and 3.3 are also added in the appendix.
>
> ### Q5. Writing: please revise the wording in this paper to be more objective. For example, the word "you" or "your" appear 6 times in total.
>
> A5. Thank you for these suggestions. The authors will make the paper more objective. The usage of the word "you" and "your" have been removed from the paper.
>
> ---
> ### References
> Vikash Sehwag, Mung Chiang, and Prateek Mittal. Ssd: A unified framework for self-supervised outlier detection. arXiv preprint arXiv:2103.12051, 2021.

---

> > ### Comment · Reviewer_yEDC · 2024-11-22
> >
> > Thanks for your response. The contributions are clearer to me now. I would think of raising the score but there is no option for 7 while I don't think the contributions are as impressive as to rate 8. 7 would be my rating.

---

> ### Author Response · Authors · 2024-11-22
>
> Thank you very much for reading our response! We are happy to know that our answers and revised paper have made the contributions more clear. We also appreciate your willingness to increase the rating and support of our work!
>
> Best,
>
> Authors

---

### Official Review · Reviewer_kvex · 2024-10-30

**Soundness:** 3
**Presentation:** 2
**Contribution:** 3
**Rating:** 5
**Confidence:** 4

**Summary:**

This paper mainly focuses on the Out-of-distribution (OOD) detection problem in self-supervised learning settings, which investigates the conditions for the theoretical guarantee of failure in unlabeled OOD detection from the information-theoretic perspective. In summary, this work first identifies one scenario in which any self-supervised learning or unsupervised learning algorithm will fail when there exists zero mutual information between the learning objective and in-distribution labels and then introduces a new detection task, namely adjacent OOD detection, which is theoretically proved to possibly exist in real-world dataset. Experiments with several existing self-supervised and unsupervised learning OOD methods are conducted to support the major claims of this work.

**Strengths:**

1. This paper focuses on a new setting, i.e., self-supervised learning OOD, and investigates the feasibility of OOD detection from a theoretical perspective.
2. This paper theoretically proved the existence of failure conditions for unlabeled OOD detection algorithms, namely, label blindness, providing insights from the information-theoretic perspective.
3. This paper also introduces a new task, i.e., adjacent OOD detection, which is theoretically demonstrated to be possible in the real-world dataset.

**Weaknesses:**

1. The writing and presentation can be further improved, since there are numerous typos and unclear notation or concept definitions. To list some, there is no clear introduction to the general paradigm of unlabeled OOD detection algorithms, and the preliminary about OOD detection is unclear in how to detect the OOD sample beyond using $P(y\notin Y_{in}|x)$.
2. The motivation and practical significance for considering self-supervised OOD detection is not clear. Given the current presentation and introduction content, the reviewer can hardly understand why we should conduct unlabeled OOD detection. If we have the pre-trained model on the ID data, even if we adopt the self-supervised learning in pertaining, can we have the pseudo label for those unlabeled data for OOD detection? It could be better to illustrate the overall research setting using some figure or framework before the existing Figure 1 for a better explanation.
3. It seems there are no explicit assumptions for the theoretical results and most theoretical results are based on the sufficiency definition in definition 2.1 and the minimal sufficient statistic in definition 2.2. It is questionable whether the theoretical insights reflect the real-world challenge or not, as there is limited empirical justification for those intermediate theoretical results based on mutual information. It could be better if the presentation incorporated more quantitative results regarding the mutual information.

**Questions:**

1. It seems to be an unknown reference at the end of the first key contributions on Page 2.
2. Although the adjacent ood is discussed to be different from those previously defined near ood or far ood scenarios, what is the difference between adjacent ood with anonymous outlier detection if there is significant feature overlap at both settings?

---

> ### Author Response · Authors · 2024-11-22
> **Response to Reviewer 2**
>
> Thank you for your thoughtful and constructive feedback. We address your comments and questions below.
> ### Q1. The writing and presentation can be further improved, since there are numerous typos and unclear notation or concept definitions. To list some, there is no clear introduction to the general paradigm of unlabeled OOD detection algorithms, and the preliminary about OOD detection is unclear in how to detect the OOD sample beyond using $P(y \notin Y_{in} | x)$
>
> A1. We have updated the paper to remove typos and improve definitions. Regarding the introduction of unlabeled OOD detection, the authors acknowledge that the previous version lacked any specific description in the preliminaries section of the paper. We have added a paragraph in Section 2 explaining how unlabeled OOD differs from labeled OOD. In Section 2, we have also added a description of the baseline OOD detector, maximum softmax probability. We agree that providing an example OOD detector is very important in that section.
> ### Q2. The motivation and practical significance for considering self-supervised OOD detection is not clear. Given the current presentation and introduction content, the reviewer can hardly understand why we should conduct unlabeled OOD detection.
>
> A2. We have updated the introduction to explain why one might wish to pursue unlabeled OOD detection.  Some of the reasons include saving costs on labeling and allowing a model to scale to extremely large unlabeled datasets. There is also growing interest in training large self-supervised models that do not rely on labels, such as unconditional diffusion models. Building such models with robust OOD detection functionality could greatly improve their efficacy in real-world, especially safety-critical situations. Given these reasons, a good number of self-supervised OOD detection methods (see the references in the second paragraph of the introduction section) have emerged with promising results.
>
> ### Q3. If we have the pre-trained model on the ID data, even if we adopt the self-supervised learning in pertaining, can we have the pseudo label for those unlabeled data for OOD detection?
>
> A3. Assuming that a pseudo labeling process would involve some initial data that is labeled, such a process would not be considered unlabeled and beyond the scope of this work. Further work would be needed to investigate that scenario. Generally, adding labels will improve OOD detection performance and it is possible to use a mix of labeled and unlabeled data. See paper (Du et al 2024)
>
>
> ### Q4. It could be better to illustrate the overall research setting using some figure or framework before the existing Figure 1 for a better explanation.
> A4. Thank you for the suggestion. We have added a figure (Figure 1 on page 2 of the revised paper) to provide a visualization of the unlabeled and labeled OOD processes.
>
> ### Q5. It seems there are no explicit assumptions for the theoretical results and most theoretical results are based on the sufficiency definition in definition 2.1 and the minimal sufficient statistic in definition 2.2. It is questionable whether the theoretical insights reflect the real-world challenge or not, as there is limited empirical justification for those intermediate theoretical results based on mutual information. It could be better if the presentation incorporated more quantitative results regarding the mutual information.
>
> A5. There are notable existing works that study the  information bottleneck effect of deep learning models by investigating mutual information between the input, model learned representations, and the output. For example, the work by (Federici et al., 2020) theoretically demonstrates the information bottleneck effect, the minimal sufficient statistic, and how mutual information affects neural networks. The theories are empirically verified through comprehensive quantitative results, where the reviewer can refer to.
>
> To better connect our theoretical insights with real-world challenges, we have revised the paper to propose two concepts in Section 3.1: **strict label blindness** and **approximate label blindness**. Strict label blindness is equivalent to the label blindness proposed in the earlier draft. Approximate label blindness describes situations where the mutual information between the learning objective and labels is close to zero. We utilize Fano’s inequality to establish a lower bound for classification error based on the mutual information content, such that low mutual information leads to a high lower bound for the prediction error. Under the approximate label blindness scenario, we expect the OOD algorithm to behave almost as poorly as in the strict label blindness scenario. Our empirical results evaluate the approximate label blindness condition, which should be more relevant for real world situations.
>
> *Response continues in the following comment.*

---

> ### Author Response · Authors · 2024-11-22
> **Response to Reviewer 2 Continued**
>
> ### Q6. It seems to be an unknown reference at the end of the first key contributions on Page 2.
> A6. This has been fixed.
> ### Q7. Although the adjacent ood is discussed to be different from those previously defined near ood or far ood scenarios, what is the difference between adjacent ood with anonymous outlier detection if there is significant feature overlap at both settings?
> A7. Our understanding is that anonymous outlier detection is the same as unsupervised outlier detection. If this is incorrect, please let us know.
>
> With regard to anonymous/unsupervised outlier detection, a major difference is that the adjacent OOD benchmark does not interpret outliers in a comparable way. For example, given classes $Y = [1, 2, 3]$ and data points generated as $X = N(Y, 1)$, where $N$ is a normal distribution with $N(\mu, \sigma)$, it is unlikely that an unsupervised outlier detection method would consider data points labeled $Y = 2$ as outliers (since these data points would be caught in the middle of the distributions of $X$ with label $Y = 1$ and $Y = 3$). However, the adjacent OOD detection benchmark would test each class as OOD, with the others as ID. This means that the data labeled as $Y = 2$ would be OOD and $Y = [1,3]$  would be ID, which would test $Y = 2$ as out of distribution. In a way, the adjacent OOD benchmark considers a whole class as a form of outlier and tests the model’s ability to separate that outlier from the other known classes.
>
> ---
> ### References
>
> Xuefeng Du, Yiyou Sun, and Yixuan Li. When and how does in-distribution label help out-of-distribution detection? arXiv preprint arXiv:2405.18635, 2024
>
> Marco Federici, Anjan Dutta, Patrick Forr´e, Nate Kushman, and Zeynep Akata. Learning robust representations via multi-view information bottleneck. arXiv preprint arXiv:2002.07017, 2020.

---

> ### Comment · Area_Chair_dwN9 · 2024-11-30
>
> Dear Reviewer kvex,
>
> Many thanks for reviewing this paper. Could you have a look at the authors' responses and provide further comments if any?
>
> Best,
>
> Your Area Chair

---

### Official Review · Reviewer_EZPK · 2024-11-04

**Soundness:** 4
**Presentation:** 3
**Contribution:** 2
**Rating:** 8
**Confidence:** 3

**Summary:**

The paper addresses challenges in out-of-distribution (OOD) detection, focusing on "label blindness," where OOD algorithms fail due to a lack of labeled data. It proves that unlabeled OOD detection inherently fails when label information is not linked to the learning objective, proposing an Adjacent OOD benchmark to capture these failures. Experimental results reveal that existing unlabeled methods struggle with overlapping IID and OOD data.

**Strengths:**

- theoretical foundation proving that unlabeled OOD detection methods are inherently limited by "label blindness" -- a fundamental flaw in current self-supervised and unsupervised OOD approaches.
- novel OOD detection benchmark to test OOD methods under conditions of high overlap between in- and out-of-distribution data, filling a critical gap in safety evaluation.
- Extensive experiments that demonstrate current unlabeled OOD methods often fail in real-world scenarios with overlapping ID and OOD data, highlighting the need for label-aware approaches.
- Offers clear guidance for future OOD research by identifying key challenges and proposing benchmarks that better capture the limitations of current methods.

**Weaknesses:**

- Limited applicability of theoretical claims -- while the paper introduces a "Label Blindness Theorem," rely heavily on specific assumptions about mutual information and independence (e.g., Theorem 3.1 and Corollary 3.3), which may limit the generalizability of the "label blindness" concept to real-world applications where these assumptions don't hold.
- the practical applicability of this theory is not fully explored and assume idealized conditions (Section 3.1) which might not reflect the complexity of real-world OOD scenarios.
- Relies heavily on specific datasets like ICML Facial Expressions, Stanford Cars, and Food-101, which limits generalizability, as these may not capture the diversity of OOD detection in other modalities such as language or video or audio.

**Questions:**

- Could the benchmark be vulnerable to artificially low variance during testing due to potential biases in data selection or label overlap?
- It seems that the notion of of "label blindness" is central to the theory of this paper. How do you address practical situations where correlations between features and labels might enable OOD detection without explicit labels, and would this differ from the guarantees provided by you theoretical assumptions?

---

> ### Author Response · Authors · 2024-11-22
> **Response to Reviewer 1**
>
> Thank you for your thoughtful and constructive feedback. We address your comments and questions below.
>
>
>
> ### Q1. Limited applicability of theoretical claims -- while the paper introduces a "Label Blindness Theorem," rely heavily on specific assumptions about mutual information and independence (e.g., Theorem 3.1 and Corollary 3.3), which may limit the generalizability of the "label blindness" concept to real-world applications where these assumptions don't hold.
>
> A1. We agree that the most strict interpretation of the theory could be limiting as there may not be many situations where the mutual information between two real world variables is exactly zero. We have revised the paper to propose two concepts in Section 3.1: strict label blindness and approximate label blindness. Strict label blindness is equivalent to the label blindness proposed in the earlier draft. Approximate label blindness describes situations where the mutual information between the learning objective and labels is close to zero. We utilize Fano’s inequality to establish a lower bound for classification error based on the mutual information content, such that low mutual information leads to a high lower bound for the prediction error. Under the approximate label blindness scenario, we expect the OOD algorithm to behave almost as poorly as in the strict label blindness scenario. Our empirical results evaluate the approximate label blindness condition, which should be more relevant for real world situations.
>
> ### Q2. the practical applicability of this theory is not fully explored and assume idealized conditions (Section 3.1) which might not reflect the complexity of real-world OOD scenarios.
>
> A2. In the previous version of the paper, there was not enough emphasis on how the theory would behave in complex real world scenarios. We added additional discussion in Section 3.1 regarding how the approximate label blindness theory can be used in real world applications to determine when unlabeled OOD methods may be appropriate. We also added a distinction between strict and approximate label blindness, described in A1.
>
> ### Q3. Relies heavily on specific datasets like ICML Facial Expressions, Stanford Cars, and Food-101, which limits generalizability, as these may not capture the diversity of OOD detection in other modalities such as language or video or audio.
>
> A3. The authors agree that gathering empirical evidence in other modalities would be worthwhile. We would like to clarify that our theoretical contribution is independent of the modality and can be generally applied to different data modalities. The authors chose to focus their empirical efforts on images due to the large amount of previous work on images in OOD detection. Applying the theories to other data modalities as suggested by the reviewer is an interesting future direction.
>
> ### Q4. Could the benchmark be vulnerable to artificially low variance during testing due to potential biases in data selection or label overlap?
>
> A4. If variance is referred to as the differences between images (input data) of different labels, then we can consider this as similar to the covariance of the images and labels (images of label A would have a different distribution than images of label B). If we assume that some non random function was used to generate the labels (eg. human labeling), then there must exist variance between images of different labels.
>
> During the learning process, the images are processed by a model into intermediate representations. However, these representations may not exhibit any covariance with the labels. For example, if the distribution of representations for label A and label B are the same, then it would be impossible for the model to detect those labels using said representations. This failure state is specifically what the benchmark is testing for, i.e., the benchmark aims to detect artificially low variance in the model’s learned representations.
>
> *Response continues in the following comment.*

---

> ### Author Response · Authors · 2024-11-22
> **Response to Reviewer 1 Continued**
>
> ### Q5. It seems that the notion of of "label blindness" is central to the theory of this paper. How do you address practical situations where correlations between features and labels might enable OOD detection without explicit labels, and would this differ from the guarantees provided by your theoretical assumptions?
>
> A5. There are existing methods that perform unlabeled OOD detection, which learn features that are correlated with the underlying labels, while still not accessing the labels. This includes the SimCLR based method SSD (Sehwag et al 2021) and others referenced in this paper. This paper tests those methods on datasets where label blindness occurs. The guarantee of complete failure only occurs when the assumption holds. When the assumption is approximate (mutual information is close to zero), we can still guarantee a large degree of failure because the effectiveness of the algorithm is limited by the upper bound defined by the mutual information value, as shown in the revised section 3.1.
>
> We added a Section 7.2 in the revised paper to discuss the impact of label blindness on real-world problems.  In particular, unlabeled OOD can also be used in cases where the risk of adjacent OOD data is acceptably low. The risk defined by Theorem 4.1 is less relevant when randomness in the collection of ID data can be ensured. For example, an ID dataset of World War 2 aircraft would not be biased by the collection date and the risk of overlapping OOD data can be reduced to effectively zero. Unlabeled OOD detection can also work well when the learned features are relevant for the OOD setting. In the case of adjacent OOD detection, an unlabeled method should perform well if the learning objective is closely related to the ID labels. Alternatively, unlabeled OOD detection can be used in cases where one expects only near and far OOD data.
>
> ---
> ### References
>
> Vikash Sehwag, Mung Chiang, and Prateek Mittal. Ssd: A unified framework for self-supervised outlier detection. arXiv preprint arXiv:2103.12051, 2021.

---

> > ### Comment · Reviewer_EZPK · 2024-11-25
> >
> > Thank you for addressing my concerns. I've now raised my rating to an 8.

---

> > > ### Author Response · Authors · 2024-11-25
> > >
> > > It is great to know that our response has adequately addressed your concerns. We highly appreciate your support of our work by increasing the score!
> > >
> > > Best,
> > >
> > > Authors

---

### Comment · Area_Chair_dwN9 · 2024-11-22

Dear Authors and Reviewers,

The discussion phase has passed 10 days. If you want to discuss this with each other, please post your thoughts by adding official comments.

Thanks for your efforts and contributions to ICLR 2025.

Best regards,

Your Area Chair

---

### Author Response · Authors · 2024-11-22
**General Response**

The authors would like to thank all reviewers for taking the time to provide insightful feedback. In this general response, we would like to highlight our answers to some key questions raised by reviewers:

### Q1. Reviewers questioned the general applicability of theoretical claims due to the strict nature of the assumptions (eg. heavy reliance on independence and mutual information). The reviewers questioned if the assumptions would hold in real world applications (from **Reviewers  EZPK, kvex**).

A1. The authors have extended the theory using Fano’s Inequality to consider cases where the learning objective and the labels are almost independent. Fano’s Inequality guarantees a lower bound on the prediction error, where the lower bound increases with a decrease in the mutual information between the learning objective and labels. This implies that when there is little mutual information, the lower bound of prediction error is extremely high.

Such cases are now described as approximate label blindness to distinguish them from strict label blindness in the revised Section 3.1. The conditions in approximate label blindness better reflect the experiments conducted in this paper, where there is minimal mutual information between the learning objectives and labels. Through Fano’s inequality, we can apply the label blindness theorem to more real-world applications.

The authors note that the experiments in this paper are conducted with the approximate label blindness settings, as there still exists some mutual information between the learning objective and labels in the datasets used in the benchmark. The approximate label blindness conditions may apply to many datasets and unlabeled OOD methods.

### Q2. Reviewers questioned the impact of this paper on future research in unlabeled OOD detection and its applications. The reviewers noted that this paper identifies a theoretical problem present in unlabeled OOD detection, but does not provide a solution (from **Reviewers  EZPK, yEDC**).

A2. The authors have improved the discussion section to provide guidance on how unlabeled OOD research can be improved. One method is to develop both labeled and unlabeled methods together, as was done in (Sehwag et al 2021). This allows the labeled alternative to be deployed where the unlabeled option cannot.

The authors also introduce discussion on the conditions in which one can safely deploy unlabeled OOD detection systems. Generally, these are cases where the assumptions of the theory do not hold (the learning objective and labels are strongly correlated) or when the risk of adjacent OOD data is sufficiently low or tolerable (if one could guarantee only near and far OOD detection).

The paper has been updated to capture all the changes mentioned in our general response and detailed rebuttal to each review, where the changes are highlighted in blue.

---
### References
Vikash Sehwag, Mung Chiang, and Prateek Mittal. Ssd: A unified framework for self-supervised outlier detection. arXiv preprint arXiv:2103.12051, 2021.

---

### Meta-Review · Area_Chair_dwN9 · 2024-12-21

**Metareview:**

Mainstream research in OOD detection focuses on classification problems that require a large amount of labeled data. Obtaining labeled data is a well-known burden for small companies or universities. Thus, this paper focuses on OOD detection in a self-supervised learning setting where is more often considered in the real world. The paper provides an essential analysis of this problem and proposes a novel algorithm in the end. The performance is validated via experiments as well.

Only one reviewer questions the significance of this problem. However, after checking the authors' responses and the paper itself, the studied problem is interesting and has a certain value to the field. The authors should merge all comments into the next version of this paper.

**Additional Comments On Reviewer Discussion:**

Only one reviewer questions the significance of this problem. However, after checking the authors' responses and the paper itself, the studied problem is interesting and has a certain value to the field.

---

### Decision · Program_Chairs · 2025-01-22

Accept (Poster)